# A Close Look at Negative Label Guided Out-of-distribution Detection in Pre-trained Vision-Language Models

**Bo Peng** [1] **Jie Lu** [1] **Zhen Fang** [1] **Guangquan Zhang** [1]

## Abstract

Advances in pre-trained vision-language models have enabled zero-shot out-of-distribution (OOD) detection using only in-distribution (ID) labels. Recent methods in this direction expand the label space with negative labels to enhance the discrimination between ID and OOD inputs. Despite their promising progress, there remains a limited understanding of their empirical effectiveness in open-world scenarios, where negative labels can arbitrarily diverge from real OOD ones. This paper bridges this research gap with the helm of a novel energy-based framework, where the energy function is built upon the margin between the similarity of an input to ID labels and that to negative labels. Guided by this framework, we prove that the inherent tolerance of such methods to the sampling bias essentially stems from estimating the worst-case energy function over a KL-constrained set of potential distributions centered on the negative label distribution. Furthermore, our theoretical analysis reveals that existing methods suffer from over-pessimism and consequently high sensitivity to outliers. Provably, we can alleviate these problems by leveraging Rényi divergence to refine potential distributions. Extensive experiments empirically manifest that our method establishes a new state-of-the-art across a variety of OOD detection settings. Code is publicly available at here.

## 1. Introduction

Despite the significant progress in machine learning that has facilitated a broad spectrum of classification tasks (Masana et al., 2022; Zhao et al., 2019; Caruana & Niculescu-Mizil, 2006; Peng et al., 2020; Zhu et al., 2020; 2023), models often operate under a *closed-world* scenario, where test data stems from the same distribution as the training data. However, real-world applications often entail *open-world* scenarios in which deployed models may encounter unseen classes of data during training, giving rise to what is known as out-of-distribution (OOD) data. These OOD data can potentially undermine a model's stability and, in certain cases, inflict severe damage on its performance. Accordingly, a reliable discriminative model should not only correctly classify known in-distribution (ID) data but also flag any OOD data as unknown. This directly motivates OOD detection (Lang et al., 2023; Salehi et al., 2021; Yang et al., 2021), which makes significant differences in ensuring the safety of decision-critical applications, e.g., autonomous driving (Huang et al., 2020), medical diagnosis (Zimmerer et al., 2022), and cyber-security (Nguyen et al., 2022).

This paper focuses on post-hoc OOD detection, which is more practical than learning-based methods that require resource-intensive retraining. Earlier studies (Liang et al., 2017; Liu et al., 2020; Huang et al., 2021; Sun et al., 2022) primarily utilized the single modality of pre-trained models, but the success of contrastive language-image pre-training (CLIP) (Radford et al., 2021a) has recently shifted research toward expanding post-hoc OOD detection from single-modal to multi-modal methods. The pioneering work, MCM (Ming et al., 2022), defines textual features as the concept for each ID class and uses the scaled distance between visual features and the closest ID prototype to measure OOD uncertainty. This method has paved the way for using pre-trained vision-language models (VLMs) in post-hoc OOD detection. However, MCM relies only on textual information from the ID label space, leaving the text interpretation capabilities of VLMs underutilized. To address this, NegLabel (Jiang et al., 2024) selects negative labels from large-scale lexical databases, such as WordNet (Miller, 1995), based on their similarities to the ID label space, which equips the model with stronger ability to distinguish OOD data. Despite promising potential of negative labels, there remains a limited theoretical understanding of their effectiveness in open-world scenarios, where real OOD data, due to its open-ended nature, can be arbitrarily different from the observed negative labels (Wang et al., 2023b;c).

[1]Faculty of Engineering & Information Technology, University of Technology Sydney, Sydney, Australia. Correspondence to: Jie Lu <Jie.Lu@uts.edu.au>.

*Proceedings of the $43^{rd}$ International Conference on Machine Learning*, Seoul, South Korea. PMLR 306, 2026. Copyright 2026 by the author(s).

To mitigate this research gap, this paper delivers a close look at CLIP-based post-hoc OOD detection with negative labels from the perspective of density estimation. We argue that this standpoint is well-suited for studying OOD detection, since OOD data, by definition, diverges from ID data in terms of their underlying density distributions. Following prior works (Liu et al., 2020), our analytical framework models ID data by resorting to the energy-based model (LeCun et al., 2006). However, we find that it is non-trivial to extend the energy function from a uni-modal to a multi-modal setting. Drawing inspiration from triplet-based metric learning (Sohn, 2016; Hermans et al., 2017), we propose to build the energy function upon the margin between the similarity of a given test-time input to ID labels and to negative labels. Guided by this framework, we theoretically show that NegLabel essentially augments the negative label distribution by constructing a distribution set contained within a Kullback–Leibler (KL) ball centered on it. Estimating the energy function against the worst-case distribution in this set ensures performance guarantees under all possible (or constrained) distribution shifts. This provides a theoretical explanation for why NegLabel remains effective when faced with unseen OOD data.

At the same time, our theoretical analysis reveals that the paradigm of NegLabel is prone to induce an overly conservative worst-case distribution, as it assigns disproportionately large weights (governed by an exponential function) to negative labels that exhibit high similarity to the test-time input. In response, we transcend the boundary of KL divergence, but exploring a broader family of distribution divergence metrics — Rényi divergence (Rényi, 1961). As a generalization of KL divergence, Rényi divergence introduces an additional parameter, the order, which offers flexible control over the weighting distribution. We show the use of Rényi divergence enables to retain the aforementioned strengths while mitigating the conservativeness by shaping a milder, polynomial-bounded worst-case distribution with adaptively tunable order. Extensive experiments empirically show that our method establishes a new state of the art in a variety of OOD detection setups.

## 2. Related Work

The core of CLIP-based OOD detection lies in how to leverage texture supervision with pre-trained VLMs to assist OOD detection on the visual domain. On the one hand, the pioneering work, MCM (Ming et al., 2022), defines textual features as concept proto-types for each ID class and uses the scaled distance between visual features and the closest ID prototype to measure OOD uncertainty. Instead of relying on textual information from only ID label space, ZOC (Esmaeilpour et al., 2022) applies VLMs to discern OOD instances by training a captioner that generates poten-

tial OOD labels. Nevertheless, this captioner often fails to produce effective OOD labels, particularly for ID datasets containing many classes. Differently, NegLabel (Jiang et al., 2024) incorporates additional negative class names mined from available data sources as negative proxies. Considering the nonalignment between target visual OOD distribution and the generated negative textual OOD distribution, AdaNeg (Zhang & Zhang, 2024) leverages the benefits of test-time adaptation to generate adaptive proxies by exploring potential OOD images during testing. More recently, Peng et al. (2026b) understand CLIP-based post-hoc OOD detection from an information-theoretical perspective while Peng et al. (2026c) improve negative mining strategy in CLIP-based post-hoc OOD detection from a positive-unlabeled perspective. On the other hand, CLIP-based OOD detection can also be improved by prompt representation learning. In particular, LoCoOp (Miyai et al., 2024) learns ID text prompts by pushing them away from the portions of CLIP local features that have ID-irrelevant nuisances (e.g., backgrounds). CLIPN (Wang et al., 2023a) and LSN (Nie et al., 2024) design a learnable "no" prompt and a "no" text encoder to capture negation semantics within images. Differently, LAPT (Zhang et al., 2025) initializes prompts with negative labels (Jiang et al., 2024), followed by tuning prompts with cross-modal and cross-distribution mixing. *More related works are discussed in Appendix A.*

## 3. Preliminary

**Notation.** Let $\mathcal{X}$ and $\mathcal{Y}$ be the input space and the label space, respectively. Given a random variable $Y \in \mathcal{Y}$, we write $\mathbb{P}_Y$ as the marginal distribution defined over $\mathcal{Y}$, and use $y \sim \mathbb{P}_Y$ to indicate a sample $y$ drawn from $\mathbb{P}_Y$. Considering $K$-way classification, we write $\mathcal{Y}_\mathrm{I} \triangleq \{y_1, \ldots, y_K\} \subset \mathcal{Y}$ as the *known* ID label space. The joint ID distribution, represented as $\mathbb{P}_{X_\mathrm{I} Y_\mathrm{I}}$, is a joint distribution defined over $\mathcal{X} \times \mathcal{Y}_\mathrm{I}$. During testing, there are some unknown OOD joint distributions $\mathbb{P}_{X_\mathrm{o} Y_\mathrm{o}}$ defined over $\mathcal{X} \times \mathcal{Y}_\mathrm{o}$, where $\mathcal{Y}_\mathrm{o} \subseteq \mathcal{Y} \backslash \mathcal{Y}_\mathrm{I}$ is the *unknown* OOD label space.

**Post-hoc OOD Scoring.** Existing methods (Hendrycks & Gimpel, 2016; Liang et al., 2017; Liu et al., 2020; Huang et al., 2021; Sun et al., 2022) tend to adopt a post-hoc strategy to detect OOD data, *i.e.,* given a pre-trained ID classification model $f$ and a scoring function $S(\cdot; f) : \mathcal{X} \to \mathbb{R}$, then $\mathbf{x}$ is detected as ID data if and only if $S(\mathbf{x}; f) \geq \lambda$, for some given threshold $\lambda$:

$$g(\mathbf{x}) = \mathrm{ID}, \text{ if } S(\mathbf{x}; f) \geq \lambda; \text{ otherwise, } g(\mathbf{x}) = \mathrm{OOD}.$$

Typically, $\lambda$ is chosen to ensure a high fraction (e.g., 95%) of ID data to be correctly classified.

**CLIP-based Models** adopt a dual-stream architecture (Radford et al., 2021b) with one text encoder $f_\mathcal{T}$ and one image encoder $f_\mathcal{X}$ to map inputs of two modalities into a uni-modal

hyper-spherical space $\mathbb{S}^{d-1} \triangleq \left\{ \mathbf{z} \in \mathbb{R}^d \mid \|\mathbf{z}\|_2 = 1 \right\}$. Zero-shot image classification based on a pre-trained CLIP-like model is to classify images into one of known ID classes by computing $\arg\max_{j=1,\dots K} h(\mathbf{x}, y_j)$ where $h(\mathbf{x}, y_j) \triangleq f_{\mathcal{X}}(\mathbf{x})^\top f_{\mathcal{T}}(\Delta(y_j))$ with $\Delta(\cdot)$ producing the text prompt for the input label.

**CLIP-based OOD Detection with Negative Labels.** CLIP-based models are recently extended to the task of zero-shot OOD detection where there is no need to train on ID data. A popular pipeline is to leverage a $L$-sized set of negatives labels[1] $\hat{\mathcal{Y}} \triangleq \{\hat{y}_1, \dots, \hat{y}_L\}$ to formulate the OOD scoring function of $\mathbf{x}$ as the model's prediction confidence that $\mathbf{x}$ belongs to $\mathcal{Y}_{\mathrm{I}}$, i.e.,

$$S_{\mathrm{NegLabel}}(\mathbf{x}; f) \triangleq \frac{\sum_{i=1}^{K} e^{h(\mathbf{x}, y_i)/T}}{\sum_{j=1}^{K} e^{h(\mathbf{x}, y_j)/T} + \sum_{j=1}^{L} e^{h(\mathbf{x}, \hat{y}_j)/T}}, \quad (1)$$

where $T > 0$ is a temperature hyper-parameter.

*Due to space limitation, detailed proofs of theorems in this paper are provided in Appendix B.*

# 4. A Close Look at CLIP-based OOD Detection with Negative Labels

While NegLabel (Jiang et al., 2024) has empirically emerged to be an effective post-hoc OOD detector, there is limited prior work providing a comprehensive explanation for its efficacy from a rigorous mathematical point of view. This paper fills this research gap from the perspective of *distributionally-augmented density estimation*. Motivated by Liu et al. (2020), we consider modeling the unknown true ID density function $p_{X_{\mathrm{I}}}$ of ID input marginal distribution $\mathbb{P}_{X_{\mathrm{I}}}$ by resorting to the energy-based model (LeCun et al., 2006). In particular, let $\hat{p}_{X_{\mathrm{I}}}(\mathbf{x}; \boldsymbol{\theta})$ be an estimator of the modeled ID data density $\hat{p}_{X_{\mathrm{I}}}(\mathbf{x})$ using the pre-trained CLIP-based model parameters $\boldsymbol{\theta}$, we have:

$$\hat{p}_{X_{\mathrm{I}}}(\mathbf{x}; \boldsymbol{\theta}) = \frac{\exp[E(\mathbf{x}; \boldsymbol{\theta})]}{Z(\boldsymbol{\theta})} \propto \exp[E(\mathbf{x}; \boldsymbol{\theta})], \quad (2)$$

where $Z(\boldsymbol{\theta}) = \int \exp[E_{\boldsymbol{\theta}}(\mathbf{x}; \boldsymbol{\theta})] \, d\mathbf{x}$ is an *input-independent* normalization function with

$$E(\mathbf{x}; \boldsymbol{\theta}) = T \log \sum_{i=1}^{K} \exp[E(\mathbf{x}, y_i; \boldsymbol{\theta})/T]. \quad (3)$$

The behavior of $E(\mathbf{x}; \boldsymbol{\theta})$ is largely determined by the formulation of $E(\mathbf{x}, y_i; \boldsymbol{\theta})$. A naive choice of $E(\mathbf{x}, y_i; \boldsymbol{\theta})$ is the CLIP-based zero-shot classifier logit $h(\mathbf{x}, y_i)$, which aligns with traditional energy-based OOD detection (Liu et al., 2020). However, Table 1 shows that Energy (zero-shot)

---

[1] In accordance to Jiang et al. (2024), *negative* labels are defined as those semantically *irrelevant/dissimilar* to *all* ID labels.

achieves considerably far-from-satisfactory performance (79.57% AUROC and 82.21% FPR95 in average), which implies that it is non-trivial to extend energy function $E_{\boldsymbol{\theta}}(\mathbf{x})$ from single-modal to multi-modal settings. Let $\mathbb{P}_{\hat{Y}}$ be the sampling distribution of negative labels, drawing inspiration from triplet-based metric learning (Sohn, 2016; Hermans et al., 2017), we define $E(\mathbf{x}, y_i; \boldsymbol{\theta})$ as:

$$\begin{aligned} E(\mathbf{x}, y_i; \boldsymbol{\theta}) &\triangleq \mathbb{E}_{\hat{y} \in \mathbb{P}_{\hat{Y}}}[h(\mathbf{x}, y_i) - h(\mathbf{x}, \hat{y})] \\ &= h(\mathbf{x}, y_i) - \mathbb{E}_{\hat{y} \in \mathbb{P}_{\hat{Y}}}[h(\mathbf{x}, \hat{y})], \end{aligned} \quad (4)$$

where the expectation can be effectively estimated using the observed negative labels $\hat{\mathcal{Y}}$. Table 1 shows that Eq. (4) achieves significantly better performance (93.65% AUROC and 28.82% FPR95 in average) than Energy (zero-shot), which empirically validates our design. Intuitively, if negative labels are semantically similar to unseen ground-truth OOD labels, Eq. (4) will perform well when facing real OOD data. However, the two kinds of labels could be arbitrarily distinct from each other in practice (Wang et al., 2023b;c), posing us to suspect that the power of negative labels in Eq. (4) has yet to be fully unleashed.

Regarding this, we extend the formulation of $E(\mathbf{x}, y_i; \boldsymbol{\theta})$ in Eq. (4) beyond the given distribution $\mathbb{P}_{\hat{Y}}$ to a broader family of potential distributions with perturbations. To be specific, we are interested in the worst case of $E(\mathbf{x}, y_i; \boldsymbol{\theta})$ in Eq. (4) over a set of potential distributions $\mathbb{Q}_{\hat{Y}}$, which are centered on $\mathbb{P}_{\hat{Y}}$ and constrained by a metric function $D(\mathbb{Q}_{\hat{Y}} \| \mathbb{P}_{\hat{Y}})$ within a radius $\eta > 0$, i.e.,

$$\begin{aligned} \hat{E}(\mathbf{x}, y_i; \boldsymbol{\theta}) &= h(\mathbf{x}, y_i) - \max_{\mathbb{Q}_{\hat{Y}}} \mathbb{E}_{\hat{y} \sim \mathbb{Q}_{\hat{Y}}}[h(\mathbf{x}, \hat{y})] \\ &\text{subject to} \quad D(\mathbb{Q}_{\hat{Y}} \| \mathbb{P}_{\hat{Y}}) \leq \eta, \end{aligned} \quad (5)$$

where $D(\mathbb{Q} \| \mathbb{P})$ measures the distribution discrepancy between $\mathbb{Q}$ and $\mathbb{P}$. Intuitively, $\mathbb{Q}_{\hat{Y}}$ functions as an adversary that probes the most challenging negative-label distribution. As a result, $\hat{E}(\mathbf{x}, y_i; \boldsymbol{\theta})$ is optimized against worst-case negatives, making it more conservative. This conservatism, in turn, improves reliability when the true OOD labels deviate from the observed negative labels.

**Theorem 4.1.** *By choosing $D(\cdot \| \cdot)$ as KL divergence, i.e., $D(\mathbb{Q}_{\hat{Y}} \| \mathbb{P}_{\hat{Y}}) = \int q_{\hat{Y}}(y) \log \frac{q_{\hat{Y}}(y)}{p_{\hat{Y}}(y)} dy$, we can rewrite $\hat{E}(\mathbf{x}, y_i; \boldsymbol{\theta})$ in Eq. (5) as follows:*

$$\hat{E}(\mathbf{x}, y_i; \boldsymbol{\theta})$$
$$= \alpha^*(\mathbf{x}, \mathbb{P}_{\hat{Y}}) \log \frac{e^{h(\mathbf{x}, y_i)/\alpha^*(\mathbf{x}, \mathbb{P}_{\hat{Y}})}}{\mathbb{E}_{\hat{y} \sim \mathbb{P}_{\hat{Y}}}[e^{h(\mathbf{x}, \hat{y})/\alpha^*(\mathbf{x}, \mathbb{P}_{\hat{Y}})}]} - \alpha^*(\mathbf{x}, \mathbb{P}_{\hat{Y}})\eta,$$

*where*

$$\alpha^*(\mathbf{x}, \mathbb{P}_{\hat{Y}}) = \underset{\alpha \geq 0}{\arg\min} \left\{ \alpha\eta + \alpha \log \mathbb{E}_{\hat{y} \sim \mathbb{P}_{\hat{Y}}}[e^{h(\mathbf{x}, \hat{y})/\alpha}] \right\}.$$

**Theorem 4.2** (Lemma 5 of Faury et al. (2020)). *The optimal $\alpha^*(\mathbf{x}, \mathbb{P}_{\hat{Y}})$ can be approximated as*

$$\alpha^*(\mathbf{x}, \mathbb{P}_{\hat{Y}}) \approx \sqrt{\mathbb{V}_{\hat{y} \sim \mathbb{P}_{\hat{Y}}}[h(\mathbf{x}, \hat{y})]/2\eta}, \qquad (6)$$

*where $\mathbb{V}_{\hat{y} \sim \mathbb{P}_{\hat{Y}}}[h(\mathbf{x}, \hat{y})]$ denotes the variance of $h(\mathbf{x}, \hat{y})$ over the distribution $\mathbb{P}_{\hat{Y}}$.*

If the assumption of homoscedasticity (uniform variance) holds for each input $\mathbf{x} \in \mathcal{X}$ given a fixed $\mathbb{P}_{\hat{Y}}$, Theorem 4.2 implies that we can find a $\eta > 0$ to have $\alpha^*(\mathbf{x}, \mathbb{P}_{\hat{Y}}) \approx T$. Combining this with Theorem 4.1 implies that we can approximate the distributionally augmented energy function $\hat{E}(\mathbf{x}; \boldsymbol{\theta}) = T \log \sum_{i=1}^{K} \exp \left[\hat{E}(\mathbf{x}, y_i; \boldsymbol{\theta})/T\right]$ as follows:

$$
\begin{aligned}
\hat{E}(\mathbf{x}; \boldsymbol{\theta}) &\approx T \log \sum_{i=1}^{K} \frac{e^{h(\mathbf{x}, y_i)/T}}{\mathbb{E}_{\hat{y} \sim \mathbb{P}_{\hat{Y}}}[e^{h(\mathbf{x}, \hat{y})/T}]} - T\eta \\
&\approx T \log \underbrace{\sum_{i=1}^{K} \frac{e^{h(\mathbf{x}, y_i)/T}}{\sum_{j=1}^{L} e^{h(\mathbf{x}, \hat{y}_j)/T}}}_{\hat{S}_{\text{NegLabel}}(\mathbf{x}; f)} + \underbrace{T \log L - T\eta}_{\text{constant}}.
\end{aligned} \qquad (7)
$$

While $\hat{S}_{\text{NegLabel}}(\mathbf{x}; f)$ differs from $S_{\text{NegLabel}}(\mathbf{x}; f)$ in that the term $\sum_{j=1}^{K} e^{h(\mathbf{x}, y_j)/T}$ is excluded in its denominator, Table 1 shows that $\hat{S}_{\text{NegLabel}}(\mathbf{x}; f)$ performs on par with $S_{\text{NegLabel}}(\mathbf{x}; f)$, which implies the functional equivalence between $\hat{S}_{\text{NegLabel}}(\mathbf{x}; f)$ and $S_{\text{NegLabel}}(\mathbf{x}; f)$. The theoretical connection between $\hat{S}_{\text{NegLabel}}(\mathbf{x}; f)$ and $S_{\text{NegLabel}}(\mathbf{x}; f)$ can be found in Appendix F.1. Since $\log(\cdot)$ is monotonically increasing, Eq. (7) admits the merit of $S_{\text{NegLabel}}(\mathbf{x}; f)$ as it is equivalent to augmenting the negative label distribution $\mathbb{P}_{\hat{Y}}$ by crafting a distribution set containing all the distributions in a KL ball centered on $\mathbb{P}_{\hat{Y}}$. This allows the energy-based density estimation via Eq. (4) to perform uniformly across various potential distributions of negative labels, thereby conferring inherent tolerance to distribution discrepancy between negative labels and real OOD labels.

Despite theoretical and empirical advantages of $\hat{E}(\mathbf{x}; \boldsymbol{\theta})$ over $E(\mathbf{x}; \boldsymbol{\theta})$, the use of KL divergence to measure distribution discrepancy suffers from being overly pessimistic. To illustrate this, let us start from exploring the worst case of negative label distribution as follows.

**Theorem 4.3.** *Let us define*

$$\mathbb{Q}_{\hat{Y}}^* = \arg \max_{\mathbb{Q}_{\hat{Y}}} \mathbb{E}_{\hat{y} \in \mathbb{Q}_{\hat{Y}}}[h(\mathbf{x}, \hat{y})] \quad \text{s.t. } D(\mathbb{Q}_{\hat{Y}} \| \mathbb{P}_{\hat{Y}}) \leq \eta.$$

*If we choose $D(\cdot \| \cdot)$ as KL divergence, then we have $q_{\hat{Y}}^*(\hat{y}) = \omega_{KL}(\mathbf{x}, \hat{y}) p_{\hat{Y}}(\hat{y})$ where*

$$
\begin{aligned}
\omega_{KL}(\mathbf{x}, \hat{y}) &\triangleq \frac{e^{h(\mathbf{x}, \hat{y})/\alpha^*(\mathbf{x}, \mathbb{P}_{\hat{Y}})}}{\mathbb{E}_{\tilde{y} \sim \mathbb{P}_{\hat{Y}}}[e^{h(\mathbf{x}, \tilde{y})/\alpha^*(\mathbf{x}, \mathbb{P}_{\hat{Y}})}]} \\
&\propto e^{h(\mathbf{x}, \hat{y})/\alpha^*(\mathbf{x}, \mathbb{P}_{\hat{Y}})}.
\end{aligned} \qquad (8)
$$

Theorem 4.3 implies that the use of KL divergence leads to assigning a weight $\omega_{KL}(\mathbf{x}, \hat{y})$ to each negative label $\hat{y} \sim \mathbb{P}_{\hat{Y}}$, with the weight $\omega_{KL}(\mathbf{x}, \hat{y})$ proportional to the exponential of the scaled cosine similarity. However, the explosive nature of the exponential function would make the resulting weight distribution tend to be highly skewed so that the worst-case expectation in Eq. (5) can be dominated by outliers, i.e., those exhibiting excessively high cosine similarity to the input $\mathbf{x}$, which could greatly degrade the ability of $\hat{E}(\mathbf{x}; \boldsymbol{\theta})$ in Eq. (2) to detect OOD inputs especially when the outliers contain false negative labels[2]. We note that our theoretical analysis is consistent with empirical observations in Table 1: $\hat{S}_{\text{NegLabel}}(\mathbf{x}; f)$ performs marginally better than Energy (Eq. (4)) in average and even worse than Energy (Eq. (4)) on Textures[3].

## 5. Methodology

Our goal is to refine the worst-case distribution, aiming to assign more reasonable weights to negative labels. To this end, we propose to use Rényi divergence, a generalization of KL divergence that is defined with an additional parameter called an order, to measure distribution discrepancy. In particular, we focus on the Cressie-Read family of Rényi divergence (Duchi & Namkoong, 2021; Rényi, 1961) due to its analytical benefits reflected by the following theorem:

**Theorem 5.1.** *By choosing $D(\cdot \| \cdot)$ as the Cressie-Read family of Rényi divergence, i.e.,*

$$D(\mathbb{Q}_{\hat{Y}} \| \mathbb{P}_{\hat{Y}}) = \int q_{\hat{Y}}(y) \phi_\gamma \left(\frac{q_{\hat{Y}}(y)}{p_{\hat{Y}}(y)}\right) dy, \qquad (9)$$

*where $\phi_\gamma(t) = \frac{1}{\gamma(\gamma-1)}(t^\gamma - \gamma t + \gamma - 1)$ with $\gamma > 1$, we can rewrite $\hat{E}(\mathbf{x}, y_i; \boldsymbol{\theta})$ in Eq. (5) as*

$$
\begin{aligned}
&\hat{E}(\mathbf{x}, y_i; \boldsymbol{\theta}) \\
&= h(\mathbf{x}, y_i) - \underbrace{\left\{c_\gamma(\eta) \mathbb{E}_{\hat{y} \sim \mathbb{P}_{\hat{Y}}}\left[\left(h(\mathbf{x}, \hat{y}) - \beta_{\mathbf{x}}^*\right)_+^{\gamma^*}\right]^{\frac{1}{\gamma^*}} + \beta_{\mathbf{x}}^*\right\}}_{\zeta(\mathbf{x}, \mathbb{P}_{\hat{Y}}, \beta_{\mathbf{x}}^*)},
\end{aligned}
$$
$$(10)$$

*where $\gamma^* = \gamma/(\gamma-1)$, $c_\gamma(\eta) = (1 + \gamma(\gamma-1)\eta)^{\frac{1}{\gamma}}$, $(a)_+ = \max\{a, 0\}$, and $\beta_{\mathbf{x}}^* = \arg \min_\beta \zeta(\mathbf{x}, \mathbb{P}_{\hat{Y}}, \beta)$.*

Note that Rényi divergence in Eq. (9) introduces an order parameter $\gamma$ to adjust the polynomial relationships of

---

[2]Prior works filter negative labels from a unlabeled wild corpus database with a cosine similarity-based strategy. However, there is no theoretical guarantees that cosine similarity could correctly capture semantic relationships so that the observed negative labels are inevitably contaminated by false negative labels.

[3]While $\hat{S}_{\text{NegLabel}}(\mathbf{x}; f)$ and $S_{\text{NegLabel}}(\mathbf{x}; f)$ can be enhanced by the grouping strategy (Jiang et al., 2024), existing works are fall short in providing theoretical justification for this heuristic trick.

*Table 1.* OOD detection on ImageNet-1K with CLIP B/16 as encoder. ↑ indicates larger values are better and vice versa. The best results in the last two columns are shown in bold.

| Dataset | iNaturalist | | Sun | | Places | | Textures | | Average | |
|---|---|---|---|---|---|---|---|---|---|---|
| Metric | AUROC↑ | FPR95↓ | AUROC↑ | FPR95↓ | AUROC↑ | FPR95↓ | AUROC↑ | FPR95↓ | AUROC↑ | FPR95↓ |
| Energy (zero-shot) | 85.09 | 81.08 | 84.24 | 79.02 | 83.38 | 75.08 | 65.56 | 93.65 | 79.57 | 82.21 |
| Energy (Eq. (4)) | 98.49 | 7.11 | 94.72 | 25.93 | 90.35 | 41.35 | 91.02 | 40.89 | 93.65 | 28.82 |
| $S_{\text{NegLabel}}(\mathbf{x}; f)$ (Eq. (1)) | 99.30 | 2.65 | 95.06 | 23.11 | 90.90 | 40.35 | 89.76 | 46.63 | 93.76 | 28.19 |
| $\hat{S}_{\text{NegLabel}}(\mathbf{x}; f)$ (Eq. (7)) | 99.29 | 2.67 | 95.02 | 23.22 | 90.93 | 40.30 | 89.85 | 46.31 | 93.77 | 28.13 |
| $S_{\text{ours}}(\mathbf{x}; f)$ (Eq. (11)) | 99.64 | 1.29 | 95.71 | 17.94 | 91.90 | 33.98 | 90.79 | 39.45 | **94.51** | **23.17** |

the probability distance measure with the probability ratio. This provides enhanced flexibility in measuring distribution discrepancy by re-framing the metric function design as a search for the optimal $\gamma$ within a narrow range. A similar spirit is also witnessed in Peng et al. (2024). Since Rényi divergence recovers KL divergence as $\gamma \to 1$ (Van Erven & Harremos, 2014), one can intuitively believe that Eq. (10) should perform at least not worse than Eq. (7).

Theorem 5.1 enables to formulate $\hat{E}(\mathbf{x}; \boldsymbol{\theta})$ in Eq. (3) under Rényi divergence as follow:

$$
\begin{aligned}
\hat{E}(\mathbf{x}; \boldsymbol{\theta}) &= T \log \sum_{i=1}^{K} \frac{\exp\left[h(\mathbf{x}, y_i)/T\right]}{\exp\left[\zeta(\mathbf{x}, \mathbb{P}_{\hat{Y}}, \beta_{\mathbf{x}}^*)/T\right]} \\
&\approx S_{\text{ours}}(\mathbf{x}; \boldsymbol{\theta}) \qquad (11) \\
&\triangleq T \log \sum_{i=1}^{K} \frac{\exp\left[h(\mathbf{x}, y_i)/T\right]}{\exp\left[\hat{\zeta}(\mathbf{x}, \hat{\mathcal{Y}}, \beta_{\mathbf{x}}^*)/T\right]},
\end{aligned}
$$

where

$$
\hat{\zeta}(\mathbf{x}, \hat{\mathcal{Y}}, \beta_{\mathbf{x}}^*) \triangleq c_\gamma(\eta) \left[ \frac{1}{L} \sum_{j=1}^{L} \left(h(\mathbf{x}, \hat{y}_j) - \beta_{\mathbf{x}}^*\right)_+^{\gamma^*} \right]^{\frac{1}{\gamma^*}} + \beta_{\mathbf{x}}^*.
$$

In realization of $S_{\text{ours}}(\mathbf{x}; \boldsymbol{\theta})$ in Eq. (11), one requires to obtain $\beta_{\mathbf{x}}^*$ via solving $\arg\min_\beta \zeta(\mathbf{x}, \mathbb{P}_{\hat{Y}}, \beta)$. While there is not a closed-form solution to $\beta_{\mathbf{x}}^*$, the convexity of $\zeta(\mathbf{x}, \mathbb{P}_{\hat{Y}}, \beta)$ with regard to $\beta$ (as proved in Appendix C) motivates us to find $\beta_{\mathbf{x}}^*$ via stochastic gradient descent (SGD) with a given learning rate $lr$, i.e.,

$$
\beta_{\mathbf{x}} \leftarrow \beta_{\mathbf{x}} - lr \cdot \frac{\partial}{\partial \beta_{\mathbf{x}}} \hat{\zeta}(\mathbf{x}, \hat{\mathcal{Y}}, \beta_{\mathbf{x}}). \qquad (12)
$$

**Theorem 5.2.** *Let us define*

$$
\mathbb{Q}_{\hat{Y}}^* = \arg\max_{\mathbb{Q}_{\hat{Y}}} \mathbb{E}_{\hat{y} \in \mathbb{Q}_{\hat{Y}}} \left[h(\mathbf{x}, \hat{y})\right] \quad \text{s.t.} \quad D(\mathbb{Q}_{\hat{Y}} \| \mathbb{P}_{\hat{Y}}) \le \eta.
$$

*If we choose $D(\cdot\|\cdot)$ as the Cressie-Read family of Rényi divergence defined in Eq. (9), then we have $q_{\hat{Y}}^*(\hat{y}) =$*

$\omega_\gamma(\mathbf{x}, \hat{y}) p_{\hat{Y}}(\hat{y})$, *where*

$$
\begin{aligned}
\omega_\gamma(\mathbf{x}, \hat{y}) &\triangleq c_\gamma(\eta) \frac{(h(\mathbf{x}, \hat{y}) - \beta_{\mathbf{x}}^*)_+^{\frac{1}{\gamma^*-1}}}{\mathbb{E}_{\tilde{y} \sim \mathbb{P}_{\hat{Y}}}\left[(h(\mathbf{x}, \tilde{y}) - \beta_{\mathbf{x}}^*)_+^{\gamma^*}\right]^{\frac{1}{\gamma^*}}} \qquad (13) \\
&\propto (h(\mathbf{x}, \hat{y}) - \beta_{\mathbf{x}}^*)_+^{\frac{1}{\gamma^*-1}}.
\end{aligned}
$$

Theorem 5.2 formally discloses why $S_{\text{ours}}(\mathbf{x}; \boldsymbol{\theta})$ in Eq. (11) can be less vulnerable to the over-pessimism issue. To be specific, it can be found that the weight $\omega_\gamma(\mathbf{x}, \hat{y})$ in Eq. (13) acts as a polynomial function, therefore being relatively milder than $\omega_{KL}(\mathbf{x}, \hat{y})$ in Eq. (8). This tempers pessimism by flattening the effect of outliers: those with high cosine similarity to the input $\mathbf{x}$ still matter, but not disproportionately. Table 1 shows that the theoretical superiority (c.f. Theorem 5.2) indeed translates into strong empirical performance, where $S_{\text{ours}}(\mathbf{x}; \boldsymbol{\theta})$ significantly outperforms $\hat{S}_{\text{NegLabel}}(\mathbf{x}; f)$ and Energy (Eq. (4)).

## 6. Experiments

**Evaluation Metric**. The performance of OOD detection is evaluated via two widely used metrics: 1) the false positive rate of OOD data is measured when the true positive rate of ID data reaches 95% (FPR95); 2) the area under the receiver operating characteristic curve (AUROC) is computed to quantify the probability of the ID case receiving a higher score than the OOD case.

**Implementation**. Unless otherwise specified, we employ CLIP-B/16 for zero-shot OOD detection. Following prior works (Jiang et al., 2024; Zhang & Zhang, 2024), we adopt the text prompt of 'The nice <label>.', and select $L = 8000$ negative labels from WordNet (Miller, 1995) using the same NegMining algorithm as NegLabel (Jiang et al., 2024). Notably, we show in Section 6.2 that our method can generalize well to various CLIP architectures and corpus sources. Regarding hyper-parameters for main results, we set $T = 0.01$, $\gamma = 1.05$ and $c_\gamma(\eta) = 1.2$. We learn each input-specific constant $\beta_{\mathbf{x}}^*$ by performing SGD for only 15 steps with learning rate $lr = 1e - 2$, which, as shown in Appendix E, results in negligible computational overhead. Note that we do not leverage the heuristic grouping strategy

*Table 2.* OOD detection results on ImageNet-1K with VIT B/16 CLIP as encoder. ↑ indicates larger values are better and vice versa. The best results in the last two columns are shown in bold.

| Dataset | iNaturalist | | Sun | | Places | | Textures | | Average | |
|---|---|---|---|---|---|---|---|---|---|---|
| Metric | AUROC↑ | FPR95↓ | AUROC↑ | FPR95↓ | AUROC↑ | FPR95↓ | AUROC↑ | FPR95↓ | AUROC↑ | FPR95↓ |
| **Methods requiring training (or fine-tuning)** | | | | | | | | | | |
| MSP | 87.44 | 58.36 | 79.73 | 73.72 | 79.67 | 74.41 | 79.69 | 71.93 | 81.63 | 69.61 |
| ODIN | 94.65 | 30.22 | 87.17 | 54.04 | 85.54 | 55.06 | 87.85 | 51.67 | 88.80 | 47.75 |
| Energy | 95.33 | 26.12 | 92.66 | 35.97 | 91.41 | 39.87 | 86.76 | 57.61 | 91.54 | 39.89 |
| GradNorm | 72.56 | 81.50 | 72.86 | 82.00 | 73.70 | 80.41 | 70.26 | 79.36 | 72.35 | 80.82 |
| ViM | 93.16 | 32.19 | 87.19 | 54.01 | 83.75 | 60.67 | 87.18 | 53.94 | 87.82 | 50.20 |
| KNN | 94.52 | 29.17 | 92.67 | 35.62 | 91.02 | 39.61 | 85.67 | 64.35 | 90.97 | 42.19 |
| VOS | 94.62 | 28.99 | 92.57 | 36.88 | 91.23 | 38.39 | 86.33 | 61.02 | 91.19 | 41.32 |
| NPOS | 96.19 | 16.58 | 90.44 | 43.77 | 89.44 | 45.27 | 88.80 | 46.12 | 91.22 | 37.93 |
| LSN | 95.83 | 21.56 | 94.35 | 26.32 | 91.25 | 34.48 | 90.42 | 38.54 | 92.96 | 30.22 |
| CLIPN | 95.27 | 23.94 | 93.93 | 26.17 | 92.28 | 33.45 | 90.93 | 40.83 | 93.10 | 31.10 |
| LoCoOp | 96.86 | 16.05 | 95.07 | 23.44 | 91.98 | 32.87 | 90.19 | 42.28 | 93.52 | 28.66 |
| LAPT | 99.63 | 1.16 | 96.01 | 19.12 | 92.01 | 33.01 | 91.06 | 40.32 | 94.68 | 23.40 |
| **Zero-Shot Training-free Methods** | | | | | | | | | | |
| Mahalanobis | 55.89 | 99.33 | 59.94 | 99.41 | 65.96 | 98.54 | 64.23 | 98.46 | 61.50 | 98.94 |
| Energy | 85.09 | 81.08 | 84.24 | 79.02 | 83.38 | 75.08 | 65.56 | 93.65 | 79.57 | 82.21 |
| ZOC | 86.09 | 87.30 | 81.20 | 81.51 | 83.39 | 73.06 | 76.46 | 98.90 | 81.79 | 85.19 |
| MCM | 94.59 | 32.20 | 92.25 | 38.80 | 90.31 | 46.20 | 86.12 | 58.50 | 90.82 | 43.93 |
| NegLabel | 99.49 | 1.91 | 95.49 | 20.53 | 91.64 | 35.59 | 90.22 | 43.56 | 94.21 | 25.40 |
| Ours | 99.64 | 1.29 | 95.71 | 17.94 | 91.90 | 33.98 | 90.79 | 39.45 | **94.51** | **23.17** |

as described in Jiang et al. (2024).

**Baselines**. We compare our method with MSP (Hendrycks & Gimpel, 2016), ODIN (Liang et al., 2017), Energy (Liu et al., 2020), Gradnorm (Huang et al., 2021), Vim (Du et al., 2022), KNN (Sun et al., 2022), VOS (Tao et al., 2023), NPOS (Wang et al., 2023a), ZOC (Esmaeilpour et al., 2022), CLIPN (Wang et al., 2023a), LoCoOp (Miyai et al., 2024), LSN (Nie et al., 2024), LAPT (Zhang et al., 2025), Mahalanobis (Lee et al., 2018), MCM (Ming et al., 2022), NegLabel (Jiang et al., 2024), AdaNeg (Zhang & Zhang, 2024) and CSP (Chen et al., 2024).

## 6.1. Main Results

Following prior work (Ming et al., 2022; Jiang et al., 2024; Chen et al., 2024; Zhang & Zhang, 2024), We evaluate our method on the ImageNet-1K benchmark (Deng et al., 2009), where the validation set of ImageNet-1K is designated as the ID dataset while iNaturalist (Van Horn et al., 2018), SUN (Xiao et al., 2010), Places365 (Zhou et al., 2017), and Textures (Cimpoi et al., 2014) are considered as OOD datasets. The methods listed in the upper section of Table 2, ranging from MSP (Hendrycks & Gimpel, 2016) to VOS (Tao et al., 2023), represent traditional visual OOD detection methods. Conversely, the methods in the lower section, extending from ZOC (Esmaeilpour et al., 2022) to NegLabel (Jiang et al., 2024), employ pre-trained VLMs like CLIP. Our method chieves the state-of-the-art on the ImageNet-1k benchmark, which highlights its supe-

rior performance in the zero-shot setting. Furthermore, our method can surpass traditional methods with a finetuned CLIP, demonstrating CLIP's strong OOD detection capabilities in zero-shot scenarios. This is because CLIP can parse images in a fine-grained manner, which is achieved through its pre-training on a large-scale image-text dataset.

## 6.2. Ablation Study

**Architectures.** In principle, our method is generic to the choice of visual encoder. We evaluate our method with different visual encoder architectures, including ViT-B/32, ViT-L/14 and ResNet-50, and report the corresponding OOD detection results in Table 3. On the one hand, the performance of OOD detection can be enhanced by more powerful visual encoders. On the other hand, our method consistently outperforms the most recent NegLabel regardless of the backbone architecture used, which implies the better generalization of our method over NegLabel.

**Input Size.** In principle, our method is generic to the input resolution. We evaluate our method with a larger input size, i.e., 336×336, and report the corresponding OOD detection results in Table 4. On the one hand, the performance of OOD detection can be enhanced by a larger input size. On the other hand, our method consistently outperforms the most recent NegLabel regardless of the input resolution, which implies the better generalization of ours over NegLabel.

**Corpus Sources.** The role of the corpus is to provide a

*Table 3.* OOD detection results with different CLIP architectures on ImageNet-1k as ID. ↑ indicates larger values are better and vice versa. The best results in the last two columns are shown in bold.

| Backbone | Method | iNaturalist | | SUN | | Places | | Textures | | Average | |
|---|---|---|---|---|---|---|---|---|---|---|---|
| | | AUROC↑ | FPR95↓ | AUROC↑ | FPR95↓ | AUROC↑ | FPR95↓ | AUROC↑ | FPR95↓ | AUROC↑ | FPR95↓ |
| ViT-B/32 | MCM | 92.68 | 40.49 | 89.95 | 47.83 | 88.10 | 51.47 | 85.98 | 60.04 | 89.96 | 49.96 |
| | NegLabel | 99.11 | 3.73 | 95.27 | 22.48 | 91.72 | 34.94 | 88.57 | 50.51 | 93.67 | 27.92 |
| | Ours | 99.47 | 2.10 | 95.60 | 19.36 | 91.94 | 32.83 | 89.62 | 44.87 | **94.16** | **24.79** |
| ViT-L/14 | MCM | 93.58 | 36.80 | 92.80 | 36.77 | 90.90 | 41.35 | 85.05 | 61.70 | 90.58 | 44.16 |
| | NegLabel | 99.53 | 1.77 | 95.63 | 22.33 | 93.01 | 32.22 | 89.71 | 42.92 | 94.47 | 24.81 |
| | Ours | 99.67 | 1.22 | 96.06 | 19.15 | 93.16 | 30.57 | 90.42 | 38.32 | **94.83** | **22.31** |
| ResNet50 | MCM | 91.88 | 42.97 | 89.31 | 52.84 | 84.12 | 65.75 | 85.55 | 62.15 | 87.71 | 55.93 |
| | NegLabel | 99.24 | 2.88 | 94.54 | 26.51 | 89.72 | 42.60 | 88.40 | 50.80 | 92.97 | 30.70 |
| | Ours | 99.54 | 1.48 | 94.61 | 24.58 | 89.69 | 41.64 | 90.23 | 42.70 | **93.52** | **27.60** |

*Table 4.* OOD detection results with different input resolution on ImageNet-1k as ID, where a VIT L/14 CLIP encoder is adopted. ↑ indicates larger values are better and vice versa. The best results in the last two columns are shown in bold.

| Resolution | Method | iNaturalist | | SUN | | Places | | Textures | | Average | |
|---|---|---|---|---|---|---|---|---|---|---|---|
| | | AUROC↑ | FPR95↓ | AUROC↑ | FPR95↓ | AUROC↑ | FPR95↓ | AUROC↑ | FPR95↓ | AUROC↑ | FPR95↓ |
| 336×336 | NegLabel | 99.71 | 1.12 | 95.68 | 21.84 | 93.15 | 31.79 | 90.55 | 40.46 | 94.77 | 23.80 |
| | Ours | 99.72 | 1.09 | 96.17 | 18.24 | 93.39 | 29.62 | 90.67 | 37.68 | **94.99** | **21.66** |

*Table 5.* OOD detection results with different corpus sources on ImageNet-1k as ID, where a VIT B/16 CLIP encoder is adopted. ↑ indicates larger values are better and vice versa. The best results in the last two columns are shown in bold.

| Corpus | Method | iNaturalist | | SUN | | Places | | Textures | | Average | |
|---|---|---|---|---|---|---|---|---|---|---|---|
| | | AUROC↑ | FPR95↓ | AUROC↑ | FPR95↓ | AUROC↑ | FPR95↓ | AUROC↑ | FPR95↓ | AUROC↑ | FPR95↓ |
| Common | NegLabel | 86.91 | 65.43 | 95.03 | 24.22 | 91.52 | 34.83 | 83.69 | 67.75 | 89.29 | 48.06 |
| | Ours | 88.77 | 57.76 | 94.46 | 25.57 | 91.70 | 34.18 | 85.15 | 60.04 | **90.02** | **44.64** |
| Part-of-Speech | NegLabel | 99.23 | 3.25 | 94.20 | 25.93 | 90.17 | 43.09 | 87.77 | 50.11 | 92.84 | 30.59 |
| | Ours | 99.42 | 2.46 | 94.82 | 23.29 | 91.75 | 39.48 | 91.59 | 41.86 | **94.40** | **26.77** |

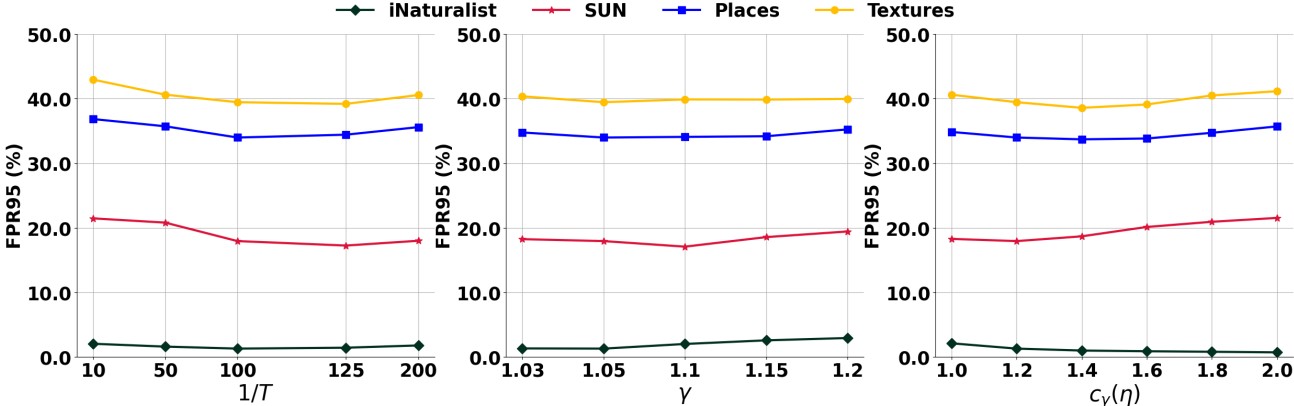

*Figure 1.* Hyper-parameter analysis on ImageNet-1K w.r.t $T$ (left), $\gamma$ (middle), and $c_\gamma(\eta)$ (right).

larger and more comprehensive semantic space. While our method is generic to the input resolution, we also conduct ablative analysis with different corpus sources, including Part-of-Speech Tags and Common-20K. As for Part-of-Speech

Tags, we, following Jiang et al. (2024), randomly sample 70000 words to constitute the corpus source. It can be found that Table 5 that our method consistently outperforms NegLabel on multiple corpora, which implies that the flexi-

*Table 6.* Evaluation on domain-generalizable OOD detection with VIT B/16 as encoder. ↑ indicates larger values are better and vice versa. The best results in the last two columns are shown in bold.

| ID Dataset | Method | iNaturalist | | SUN | | Places | | Textures | | Average | |
|---|---|---|---|---|---|---|---|---|---|---|---|
| | | AUROC↑ | FPR95↓ | AUROC↑ | FPR95↓ | AUROC↑ | FPR95↓ | AUROC↑ | FPR95↓ | AUROC↑ | FPR95↓ |
| ImageNet-S | MCM | 87.74 | 63.06 | 85.35 | 67.24 | 81.19 | 70.64 | 74.77 | 79.59 | 82.26 | 70.13 |
| | NegLabel | 99.34 | 2.24 | 94.93 | 22.73 | 90.78 | 38.62 | 89.29 | 46.10 | 93.59 | 27.42 |
| | Ours | 99.15 | 2.93 | 95.58 | 16.82 | 92.69 | 26.92 | 89.94 | 38.33 | **94.34** | **21.50** |
| ImageNet-A | MCM | 79.50 | 76.85 | 76.19 | 79.78 | 70.95 | 80.51 | 61.98 | 86.37 | 72.16 | 80.88 |
| | NegLabel | 98.80 | 4.09 | 89.83 | 44.38 | 82.88 | 60.10 | 80.25 | 64.34 | 87.94 | 43.23 |
| | Ours | 99.03 | 3.09 | 90.04 | 40.83 | 82.79 | 58.42 | 80.25 | 63.83 | **88.03** | **41.54** |
| ImageNet-R | MCM | 83.22 | 71.51 | 80.31 | 74.98 | 75.53 | 76.67 | 67.66 | 83.72 | 76.68 | 76.72 |
| | NegLabel | 99.58 | 1.60 | 96.03 | 15.77 | 91.97 | 29.48 | 90.60 | 35.67 | 94.54 | 20.63 |
| | Ours | 99.74 | 1.01 | 96.63 | 12.50 | 92.90 | 27.25 | 92.06 | 32.42 | **95.33** | **18.30** |
| ImageNetV2 | MCM | 91.79 | 45.90 | 89.88 | 50.73 | 86.52 | 56.25 | 81.51 | 69.57 | 87.43 | 55.61 |
| | NegLabel | 99.40 | 2.47 | 94.46 | 25.69 | 90.00 | 42.03 | 88.46 | 48.90 | 93.08 | 29.77 |
| | Ours | 99.64 | 1.35 | 94.72 | 23.20 | 89.93 | 42.05 | 44.77 | 89.43 | **93.44** | **27.84** |

*Table 7.* Evaluation on full-spectrum OOD detection, where a VIT B/16 CLIP encoder is adopted. ↑ indicates larger values are better and vice versa. The best results are shown in bold.

| Dataset | iNaturalist | | Sun | | Places | | Textures | | Average | |
|---|---|---|---|---|---|---|---|---|---|---|
| Metric | AUROC↑ | FPR95↓ | AUROC↑ | FPR95↓ | AUROC↑ | FPR95↓ | AUROC↑ | FPR95↓ | AUROC↑ | FPR95↓ |
| MCM | 73.97 | 84.18 | 77.97 | 78.21 | 76.02 | 78.07 | 71.82 | 81.76 | 74.94 | 80.55 |
| NegLabel | 99.38 | 2.23 | 93.97 | 28.50 | 90.58 | 41.68 | 88.84 | 51.33 | 93.19 | 30.93 |
| Ours | 99.36 | 2.32 | 94.75 | 24.15 | 91.54 | 36.34 | 88.98 | 50.00 | **93.66** | **28.20** |

bility of our method.

**Hyper-parameters.** We evaluate the hyper-parameters most essential to our algorithmic design, including the temperature $T$, the order $\gamma$, and $c_\gamma(\eta)$. The corresponding results are plotted in Figure 1.

### 6.3. Extensions

**Domain-generalizable OOD Detection.** With ImageNet-1K as a case study, we, following Jiang et al. (2024), consider ImageNet-A (Hendrycks et al., 2021b), ImageNet-R (Hendrycks et al., 2021a), ImageNet-S (Wang et al., 2019), and ImageNetV2 (Recht et al., 2019) as ID data respectively. The experiment results on four OOD datasets are shown in Table 6. The performance of MCM significantly deteriorates across diverse domain shifts, indicating the difficulty of OOD detection under such conditions. NegLabel achieves remarkably better performances than MCM, thus showing the significance of introducing negative labels for OOD detection. Our method consistently outperform NegLabel cross diverse ID datasets, which implies stronger robustness of our method against domain shifts.

**Full-spectrum OOD Detection.** We consider full-spectrum OOD detection (Yang et al., 2023), a more realistic problem setting that considers both detecting semantic shift and be-

ing tolerant to covariate shift. Different from standard and domain-generalizable settings, full-spectrum OOD detection is featured with a more fine-grained categorization of ID distributions (e.g., testing and domain-shifted ID). we, following (Zhang et al., 2023), use a mixture of ImageNet-1K validating data, ImageNet-S, ImageNet-V2, and ImageNet-C (Hendrycks & Dietterich, 2019) as ID data. The experiment results on four OOD datasets are shown in Table 7.

## 7. Conclusion

This work presents a distributionally augmented energy-based framework that offers a novel perspective on CLIP-based OOD detection with negative labels. We show that existing methods implicitly estimate the energy function against a worst-case distribution within a KL-divergence ball, which provides tolerance to sampling bias but also induces excessive pessimism. To address this limitation, we propose a Rényi-divergence–based refinement that yields a more flexible and balanced worst-case distribution, achieving state-of-the-art performance on various setups. In future work, it would be interesting to explore alternative distributional uncertainty sets, e.g., other $f$-divergences or Wasserstein metrics, to further improve robustness to semantic mismatch between negative labels and true OOD data.

**Limitation.** Our framework only provides a theoretical explanation for CLIP-based OOD detection with negative labels, leaving a crucial gap in the success of methods in other directions.

## Acknowledgment

This work was supported by the ARC Australian Laureate Fellowship (FL190100149).

## Impact Statement

Our study relies solely on publicly available datasets and models. No private or personally identifiable information was used. The work aims to advance the scientific understanding of OOD detection while upholding principles of transparency, fairness, and responsible research.

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

# A. More Related Work

## A.1. Traditional OOD Detection

The popularity of OOD detection is motivated by the empirical observation (Nguyen et al., 2015) that neural networks tend to be over-confident in OOD data. One line of work performs OOD detection by devising post-hoc scoring functions, including confidence-based methods (Hendrycks et al., 2019; Ming et al., 2022; Zhang & Xiang, 2023), energy-based methods (Liu et al., 2020), distance-based approaches (Lee et al., 2018; Sun et al., 2022; Sohn, 2016; Morteza & Li, 2022; Peng et al., 2024), gradient-based approaches (Huang et al., 2021), and Bayesian approaches (Kristiadi et al., 2020; Malinin & Gales, 2019). Another line of work addresses OOD detection by fine-tuning a pre-trained discrimination model with training-time regularizations that help the model learn ID/OOD discrepancy following the guideline of outlier exposure (Hendrycks et al., 2018). For instance, the discriminative model is regularized to produce lower confidence (Lee et al., 2017; Malinin & Gales, 2018), smaller feature magnitudes (Liu et al., 2020) or higher energy (Dhamija et al., 2018) for outlier points. More recently, some works have considered a practical scenario where the auxiliary outliers can be arbitrarily different from the real OOD data, therefore distributionally augmenting the observed OOD data. Besides, the given OOD samples tend to include unlabelled ID counterparts (Katz-Samuels et al., 2022). Because of this, WOOD (Katz-Samuels et al., 2022) formulates learning with noisy OOD samples as a constrained optimization problem while SAL (Du et al., 2024) separates candidate outliers from the unlabeled data and then trains a binary classifier using the candidate outliers and the labeled ID data.

## A.2. Vision-language Models

Pretraining on large-scale image–text pairs has established vision–language models (VLMs) as a standard backbone for multimodal transfer. Architecturally, existing VLMs can broadly be grouped into two categories: (i) *single-stream* models, which feed concatenated visual and textual features into a unified transformer, such as VisualBERT (Li et al., 2019) and ViLT (Kim et al., 2021); and (ii) *dual-stream* models, which maintain separate visual and textual encoders and learn cross-modal alignment through contrastive image–text pairing, such as CLIP (Radford et al., 2021a), ALIGN (Jia et al., 2021), SigLIP (Zhai et al., 2023), and FILIP (Yao et al., 2021). Among these, CLIP-based models have been widely adopted and have inspired numerous follow-up studies aimed at improving data efficiency and downstream adaptation (Yu et al., 2026c; 2020; 2025; 2026a;b; Liao et al., 2026; Peng et al., 2025; 2026a). In this paper, we adopt CLIP as the pretrained backbone; however, our method is generally applicable to contrastive vision–language models that learn aligned visual and textual representations.

# B. Proofs of Main Theorems

## B.1. Proof of Theorem 4.3

*Proof.* We consider the following optimization problem

$$\mathbb{Q}_{\hat{Y}}^* = \arg\max_{\mathbb{Q}_{\hat{Y}}} \mathbb{E}_{\hat{y}\in\mathbb{Q}_{\hat{Y}}}\left[h(\mathbf{x},\hat{y})\right] \quad s.t. \ D_{KL}(\mathbb{Q}_{\hat{Y}}, \mathbb{P}_{\hat{Y}}) = \int q_{\hat{Y}}(\hat{y})\log\frac{q_{\hat{Y}}(\hat{y})}{p_{\hat{Y}}(\hat{y})}d\hat{y} \leq \eta.$$

Introducing multipliers $\alpha \geq 0$ for the KL constraint and $\delta$ for normalization $\int q_{\hat{Y}}(\hat{y})d\hat{y} = 1$:

$$\mathcal{L} = \int q_{\hat{Y}}(\hat{y})h(\mathbf{x},\hat{y})\,d\hat{y} + \alpha\left(\eta - \int q_{\hat{Y}}(\hat{y})\log\frac{q_{\hat{Y}}(\hat{y})}{p_{\hat{Y}}(\hat{y})}\,d\hat{y}\right) + \delta\left(1 - \int q_{\hat{Y}}(\hat{y})\,d\hat{y}\right). \tag{14}$$

Note that $\mathcal{L}$ both depend on $\mathbb{Q}_{\hat{Y}}, \mathbf{x}, \alpha, \delta$, but we suppress the dependence from the notation for simplicity.

Taking the functional derivative with respect to $q_{\hat{Y}}(\hat{y})$ gives

$$\frac{\partial\mathcal{L}}{\partial q_{\hat{Y}}(\hat{y})} = h(\mathbf{x},\hat{y}) - \alpha\left(\log\frac{q_{\hat{Y}}(\hat{y})}{p_{\hat{Y}}(\hat{y})} + 1\right) - \delta.$$

Stationarity requires $\frac{\partial\mathcal{L}}{\partial q_{\hat{Y}}(\hat{y})} = 0$, hence

$$\log\frac{q_{\hat{Y}}^*(\hat{y})}{p_{\hat{Y}}(\hat{y})} = \frac{h(\mathbf{x},\hat{y}) - \delta - \alpha}{\alpha}.$$

Exponentiating yields

$$q_{\hat{Y}}^*(\hat{y}) = p_{\hat{Y}}(\hat{y}) \exp\left(\frac{h(\mathbf{x}, \hat{y}) - \delta - \alpha}{\alpha}\right) \propto p_{\hat{Y}}(\hat{y}) \exp\left(\frac{h(\mathbf{x}, \hat{y})}{\alpha}\right).$$

Replacing $\alpha$ with the optimal $\alpha^*(\mathbf{x}, \mathbb{P}_{\hat{Y}}) = \arg\min_{\alpha \geq 0} \left\{\alpha\eta + \alpha \log \mathbb{E}_{\hat{y} \sim \mathbb{P}_{\hat{Y}}}[e^{h(\mathbf{x}, \hat{y})/\alpha}]\right\}$ yields

$$q_{\hat{Y}}^*(\hat{y}) \propto p_{\hat{Y}}(\hat{y}) \exp\left(\frac{h(\mathbf{x}, \hat{y})}{\alpha^*(\mathbf{x}, \mathbb{P}_{\hat{Y}})}\right).$$

$\square$

## B.2. Proof of Theorem 4.1

*Proof.* Let $\ell(\hat{y}) = q_{\hat{Y}}(\hat{y})/p_{\hat{Y}}(\hat{y})$ and $\varphi(a) = a\log a - a + 1$, then we have

$$\int q_{\hat{Y}}(\hat{y}) h(\mathbf{x}, \hat{y}) \, d\hat{y} = \mathbb{E}_{\mathbb{P}_{\hat{Y}}}[h(\mathbf{x}, \hat{y})\ell(\hat{y})]$$

$$\int q_{\hat{Y}}(\hat{y}) \log \frac{q_{\hat{Y}}(\hat{y})}{p_{\hat{Y}}(\hat{y})} \, d\hat{y} = \mathbb{E}_{\mathbb{P}_{\hat{Y}}}[\varphi(\ell(\hat{y}))]$$

$$\int q_{\hat{Y}}(\hat{y}) \, d\hat{y} = \mathbb{E}_{\mathbb{P}_{\hat{Y}}}[\ell(\hat{y})]$$

According to Eq. (14), we can rewrite $\hat{E}(\mathbf{x}, y_j; \boldsymbol{\theta})$ in Eq. (5) as follows:

$$\hat{E}(\mathbf{x}, y_j; \boldsymbol{\theta}) = h(\mathbf{x}, y_j) - \min_{\alpha \geq 0, \delta} \max_{\mathbb{Q}_{\hat{Y}}} \mathcal{L}$$

$$= h(\mathbf{x}, y_j) - \min_{\alpha \geq 0, \delta} \max_{\mathbb{Q}_{\hat{Y}}} \{\mathbb{E}_{\mathbb{P}_{\hat{Y}}}[h(\mathbf{x}, \hat{y})\ell(\hat{y})] - \alpha[\mathbb{E}_{\mathbb{P}_{\hat{Y}}}[\varphi(\ell(\hat{y}))] - \eta] + \delta(\mathbb{E}_{\mathbb{P}_{\hat{Y}}}[\ell(\hat{y})] - 1)\}$$

$$= h(\mathbf{x}, y_j) - \min_{\alpha \geq 0, \delta} \left\{\alpha\eta - \delta + \alpha \max_{\mathbb{Q}_{\hat{Y}}} \{\mathbb{E}_{\mathbb{P}_{\hat{Y}}}[\frac{h(\mathbf{x}, \hat{y}) + \delta}{\alpha}\ell(\hat{y}) - \varphi(\ell(\hat{y}))]\}\right\} \tag{15}$$

$$= h(\mathbf{x}, y_j) - \min_{\alpha \geq 0, \delta} \left\{\alpha\eta - \delta + \alpha\mathbb{E}_{\mathbb{P}_{\hat{Y}}}[\max_{\ell(\hat{y})}\{\frac{h(\mathbf{x}, \hat{y}) + \delta}{\alpha}\ell(\hat{y}) - \varphi(\ell(\hat{y}))\}]\right\} \tag{16}$$

$$= h(\mathbf{x}, y_j) - \min_{\alpha \geq 0, \delta} \left\{\alpha\eta - \delta + \alpha\mathbb{E}_{\mathbb{P}_{\hat{Y}}}[\varphi^*(\frac{h(\mathbf{x}, \hat{y}) + \delta}{\alpha})]\right\} \tag{17}$$

$$= h(\mathbf{x}, y_j) - \min_{\alpha \geq 0, \delta} \left\{\alpha\eta - \delta + \alpha\mathbb{E}_{\mathbb{P}_{\hat{Y}}}[e^{\frac{h(\mathbf{x}, \hat{y}) + \delta}{\alpha}} - 1]\right\} \tag{18}$$

$$= h(\mathbf{x}, y_j) - \min_{\alpha \geq 0} \left\{\alpha\eta + \alpha \log \mathbb{E}_{\mathbb{P}_{\hat{Y}}}[e^{\frac{h(\mathbf{x}, \hat{y})}{\alpha}}]\right\} \tag{19}$$

$$= \alpha^*(\mathbf{x}, \mathbb{P}_{\hat{Y}}) \log \frac{e^{h(\mathbf{x}, \hat{y})/\alpha^*(\mathbf{x}, \mathbb{P}_{\hat{Y}})}}{\mathbb{E}_{\mathbb{P}_{\hat{Y}}}[e^{h(\mathbf{x}, \hat{y})/\alpha^*(\mathbf{x}, \mathbb{P}_{\hat{Y}})}]} - \alpha^*(\mathbf{x}, \mathbb{P}_{\hat{Y}})\eta,$$

where $\alpha^*(\mathbf{x}, \mathbb{P}_{\hat{Y}}) = \arg\min_{\alpha \geq 0} \left\{\alpha\eta + \alpha \log \mathbb{E}_{\hat{y} \sim \mathbb{P}_{\hat{Y}}}[e^{h(\mathbf{x}, \hat{y})/\alpha}]\right\}$.

We kindly note that 1) Eq. (15) holds due to the strong duality (Boyd & Vandenberghe, 2004); 2) Eq. (16) is derived via a re-arrangement for optimizing over $\mathbb{P}_{\hat{Y}}$; 3) the derivation of Eq. (17) follows by Ben-Tal & Teboulle (2007); 4) Eq. (18) is established based on the definition of convex conjugate (Hiriart-Urruty & Lemaréchal, 2004), i.e., $\varphi^*(a) = e^a - 1$.

To prove Eq. (19), Fix $\alpha > 0$ and minimize Eq. (18) over $\delta$ gives

$$\min_{\delta} \left[\alpha\eta - \delta + \alpha\mathbb{E}_{P_{\hat{Y}}}[e^{(h(\mathbf{x}, \hat{y}) + \delta)/\alpha} - 1]\right].$$

Expand and separate $\delta$ gives

$$\alpha\eta - \delta + \alpha\left(e^{\delta/\alpha}\mathbb{E}_{P_{\hat{Y}}}[e^{h(\mathbf{x}, \hat{y})/\alpha}] - 1\right) = \alpha\eta - \alpha + \underbrace{(\alpha R e^{\delta/\alpha} - \delta)}_{=:g(\delta)},$$

where $R := \mathbb{E}_{P_{\hat{Y}}}[e^{h(\mathbf{x},\hat{y})/\alpha}] > 0$ (for simplicity, we omit the dependence on $\mathbf{x}$, $P_{\hat{Y}}$ and $\alpha$).

Compute the derivative of $g(\delta)$ w.r.t. $\delta$ and set it to zero:

$$g'(\delta) = Re^{\delta/\alpha} - 1 = 0 \to \delta^* = -\alpha \log R.$$

Since $g''(\delta) = \frac{1}{\alpha} Re^{\delta/\alpha} > 0$, $\delta^*$ gives the minimum, i.e.,

$$\min_{\delta} g(\delta) = g(\delta^*) = \alpha R \cdot \frac{1}{R} - (-\alpha \log R) = \alpha + \alpha \log R.$$

Therefore, for fixed $\alpha$,

$$\min_{\delta} \left[ \alpha\eta - \delta + \alpha \mathbb{E}_{P_{\hat{Y}}}[e^{(h(\mathbf{x},\hat{y})+\delta)/\alpha} - 1] \right] = \alpha\eta - \alpha + (\alpha + \alpha \log R) = \alpha\eta + \alpha \log \mathbb{E}_{P_{\hat{Y}}}[e^{h(\mathbf{x},\hat{y})/\alpha}],$$

such that

$$h(\mathbf{x}, y_j) - \min_{\alpha \geq 0, \delta} \left[ \alpha\eta - \delta + \alpha \mathbb{E}_{P_{\hat{Y}}}[e^{\frac{h(\mathbf{x},\hat{y})+\delta}{\alpha}} - 1] \right] = h(\mathbf{x}, y_j) - \min_{\alpha \geq 0} \left[ \alpha\eta + \alpha \log \mathbb{E}_{P_{\hat{Y}}}[e^{h(\mathbf{x},\hat{y})/\alpha}] \right].$$

$\square$

### B.3. Proof of Theorem 5.1

*Proof.* We consider the following optimization problem

$$\mathbb{Q}_{\hat{Y}}^* = \arg\max_{\mathbb{Q}_{\hat{Y}}} \mathbb{E}_{\hat{y} \in \mathbb{Q}_{\hat{Y}}}[h(\mathbf{x}, \hat{y})] \quad s.t. \ D_\gamma(\mathbb{Q}_{\hat{Y}}, \mathbb{P}_{\hat{Y}}) = \int q_{\hat{Y}}(\hat{y}) \phi_\gamma \left( \frac{q_{\hat{Y}}(\hat{y})}{p_{\hat{Y}}(\hat{y})} \right) d\hat{y} \leq \eta, \tag{20}$$

where $\phi_\gamma(t) = \frac{1}{\gamma(\gamma-1)}(t^\gamma - \gamma t + \gamma - 1)$ with $\gamma > 1$.

Introducing multipliers $\alpha \geq 0$ for the Rényi constraint and $\delta$ for normalization $\int q_{\hat{Y}}(\hat{y})d\hat{y} = 1$:

$$\mathcal{L}_\gamma = \int q_{\hat{Y}}(\hat{y}) h(\mathbf{x}, \hat{y}) \, d\hat{y} + \alpha \left( \eta - \int q_{\hat{Y}}(\hat{y}) \phi_\gamma \left( \frac{q_{\hat{Y}}(\hat{y})}{p_{\hat{Y}}(\hat{y})} \right) d\hat{y} \right) + \delta \left( 1 - \int q_{\hat{Y}}(\hat{y}) \, d\hat{y} \right)$$

$$= \mathbb{E}_{\mathbb{P}_{\hat{Y}}}[h(\mathbf{x}, \hat{y}) \ell(\hat{y})] + \alpha \left( \eta - \mathbb{E}_{\mathbb{P}_{\hat{Y}}}[\phi_\gamma(\ell(\hat{y}))] \right) + \delta \left( 1 - \mathbb{E}_{\mathbb{P}_{\hat{Y}}}[\ell(\hat{y})] \right), \tag{21}$$

where $\ell(\hat{y}) = q_{\hat{Y}}(\hat{y})/p_{\hat{Y}}(\hat{y})$. Note that $\mathcal{L}_\gamma$ both depend on $\mathbb{Q}_{\hat{Y}}, \mathbf{x}, \alpha, \delta$, but we suppress the dependence from the notation for simplicity.

Then solving Eq. (20) is equivalent to solving the following problem:

$$\min_{\alpha \geq 0, \delta} \max_{\mathbb{Q}_{\hat{Y}}} \mathcal{L}_\gamma$$

$$= \min_{\alpha \geq 0, \delta} \max_{\mathbb{Q}_{\hat{Y}}} \mathbb{E}_{\mathbb{P}_{\hat{Y}}}[h(\mathbf{x}, \hat{y}) \ell(\hat{y})] + \alpha \left( \eta - \mathbb{E}_{\mathbb{P}_{\hat{Y}}}[\phi_\gamma(\ell(\hat{y}))] \right) + \delta \left( 1 - \mathbb{E}_{\mathbb{P}_{\hat{Y}}}[\ell(\hat{y})] \right)$$

$$= \min_{\alpha \geq 0, \delta} \left\{ \alpha\eta + \delta + \alpha \max_{\mathbb{Q}_{\hat{Y}}} \mathbb{E}_{\mathbb{P}_{\hat{Y}}} \left[ \frac{h(\mathbf{x}, \hat{y}) - \delta}{\alpha} \ell(\hat{y}) - \phi_\gamma(\ell(\hat{y})) \right] \right\} \tag{22}$$

$$= \min_{\alpha \geq 0, \delta} \left\{ \alpha\eta + \delta + \alpha \mathbb{E}_{\mathbb{P}_{\hat{Y}}} \left[ \max_{\ell(\hat{y})} \left( \frac{h(\mathbf{x}, \hat{y}) - \delta}{\alpha} \ell(\hat{y}) - \phi_\gamma(\ell(\hat{y})) \right) \right] \right\}$$

Note that $\max_{\ell(\hat{y})} \left\{ \frac{h(\mathbf{x},\hat{y})-\delta}{\alpha} \ell(\hat{y}) - \phi_\gamma(\ell(\hat{y})) \right\} = \phi_\gamma^* \left( \frac{h(\mathbf{x},\hat{y})-\delta}{\alpha} \right)$ is the Fenchel Conjugate Function of $\phi_\gamma(\frac{h(\mathbf{x},\hat{y})-\delta}{\alpha})$, then we have $\phi_\gamma^*(a) = \frac{1}{\gamma} \left( (\gamma-1)a + 1 \right)_+^{\gamma^*} - \frac{1}{\gamma}$ with $\gamma^* = \frac{\gamma}{\gamma-1}$ . Please refer to Duchi & Namkoong (2021) for more details.

In this way, Eq. (22) can be rewritten as follows:

$$\min_{\alpha \geq 0, \delta} \left\{ \alpha\eta + \delta + \alpha \mathbb{E}_{\mathbb{P}_{\hat{Y}}} \left[ \max_{\ell(\hat{y})} \left( \frac{h(\mathbf{x}, \hat{y}) - \delta}{\alpha} \ell(\hat{y}) - \phi_\gamma(\ell(\hat{y})) \right) \right] \right\}$$

$$= \min_{\alpha \geq 0, \delta} \left\{ \alpha\eta + \delta + \alpha \mathbb{E}_{\mathbb{P}_{\hat{Y}}} \left[ \phi_\gamma^* \left( \frac{h(\mathbf{x}, \hat{y}) - \delta}{\alpha} \right) \right] \right\}. \tag{23}$$

Following Duchi & Namkoong (2021), with $\beta = \delta - \frac{\alpha}{\gamma-1}$, we can arrive at the closed-form formulation of the optimal $\alpha^*$ that minimizes Eq. (23) as follows:

$$\alpha^* = (\gamma-1)(\gamma(\gamma-1)\eta+1)^{-\frac{1}{\gamma^*}} \mathbb{E}_{\mathbb{P}_{\hat{Y}}}\left[(h(\mathbf{x},\hat{y})-\beta)_+^{\gamma^*}\right]^{\frac{1}{\gamma^*}}, \tag{24}$$

By substituting $\alpha^*$, $\beta$ and $\phi_\gamma^*\left(\frac{h(\mathbf{x},\hat{y})-\delta}{\alpha}\right)$ into Eq. (23), we have:

$$\max_{\mathbb{Q}_{\hat{Y}}:D_\gamma(\mathbb{Q}_{\hat{Y}}\|\mathbb{P}_{\hat{Y}})\leq\eta} \mathbb{E}_{\hat{y}\in\mathbb{Q}_{\hat{Y}}}[h(\mathbf{x},\hat{y})] = \min_{\alpha\geq0,\delta}\max_{\mathbb{Q}_{\hat{Y}}}\mathcal{L}_\gamma = \min_\beta\left\{c_\gamma(\eta)\mathbb{E}_{\hat{y}\sim\mathbb{P}_{\hat{Y}}}\left[(h(\mathbf{x},\hat{y})-\beta)_+^{\gamma^*}\right]^{\frac{1}{\gamma^*}}+\beta\right\}, \tag{25}$$

where $c_\gamma(\eta) = (\gamma(\gamma-1)\eta+1)^{\frac{1}{\gamma}}$, such that

$$\begin{aligned}
\hat{E}(\mathbf{x},y_j;\boldsymbol{\theta}) &= h(\mathbf{x},y_j) - \max_{\mathbb{Q}_{\hat{Y}}:D_\gamma(\mathbb{Q}_{\hat{Y}}\|\mathbb{P}_{\hat{Y}})\leq\eta}\mathbb{E}_{\hat{y}\in\mathbb{Q}_{\hat{Y}}}[h(\mathbf{x},\hat{y})] \\
&= h(\mathbf{x},y_j) - \left\{c_\gamma(\eta)\mathbb{E}_{\hat{y}\sim\mathbb{P}_{\hat{Y}}}\left[(h(\mathbf{x},\hat{y})-\beta_{\mathbf{x}}^*)_+^{\gamma^*}\right]^{\frac{1}{\gamma^*}}+\beta_{\mathbf{x}}^*\right\},
\end{aligned} \tag{26}$$

where

$$\beta_{\mathbf{x}}^* = \arg\min_\beta\left\{c_\gamma(\eta)\mathbb{E}_{\hat{y}\sim\mathbb{P}_{\hat{Y}}}\left[(h(\mathbf{x},\hat{y})-\beta)_+^{\gamma^*}\right]^{\frac{1}{\gamma^*}}+\beta\right\}. \tag{27}$$

$\square$

## B.4. Proof of Theorem 5.2

*Proof.* Taking the functional derivative of $\frac{h(\mathbf{x},\hat{y})-\delta}{\alpha}\ell(\hat{y}) - \phi_\gamma(\ell(\hat{y}))$ in Eq. (22) w.r.t. $q_{\hat{Y}}(\hat{y})$ gives

$$\frac{\partial}{\partial\ell(\hat{y})}\left\{\frac{h(\mathbf{x},\hat{y})-\delta}{\alpha}\ell(\hat{y}) - \phi_\gamma(\ell(\hat{y}))\right\} = \frac{h(\mathbf{x},\hat{y})-\delta}{\alpha} - \frac{\partial\phi_\gamma(\ell(\hat{y}))}{\partial\ell(\hat{y})} = \frac{h(\mathbf{x},\hat{y})-\delta}{\alpha} - \frac{1}{\gamma-1}\left[\ell(\hat{y})^{\gamma-1}-1\right]. \tag{28}$$

Stationarity requires

$$\frac{\partial}{\partial\ell(\hat{y})}\left\{\frac{h(\mathbf{x},\hat{y})-\delta}{\alpha}\ell(\hat{y}) - \phi_\gamma(\ell(\hat{y}))\right\} = 0,$$

hence

$$h(\mathbf{x},\hat{y})-\delta = \frac{\alpha}{\gamma-1}\left[\ell^*(\hat{y})\right]^{\gamma-1},$$

where $\ell^*(\hat{y}) = q_{\hat{Y}}^*(\hat{y})/p_{\hat{Y}}(\hat{y})$. Replacing $\alpha$ with the optimal $\alpha^*$, then we have:

$$\ell^*(\hat{y}) = c_\gamma(\eta)\frac{(h(\mathbf{x},\hat{y})-\beta_{\mathbf{x}}^*)_+^{\frac{1}{\gamma-1}}}{\mathbb{E}_{\tilde{y}\sim\mathbb{P}_{\hat{Y}}}[(h(\mathbf{x},\tilde{y})-\beta_{\mathbf{x}}^*)_+^{\gamma^*}]^{\frac{1}{\gamma}}},$$

such that

$$q_{\hat{Y}}^*(\hat{y}) = c_\gamma(\eta)\frac{(h(\mathbf{x},\hat{y})-\beta_{\mathbf{x}}^*)_+^{\frac{1}{\gamma-1}}}{\mathbb{E}_{\tilde{y}\sim\mathbb{P}_{\hat{Y}}}[(h(\mathbf{x},\tilde{y})-\beta_{\mathbf{x}}^*)_+^{\gamma^*}]^{\frac{1}{\gamma}}}p_{\hat{Y}}(\hat{y}) \propto (h(\mathbf{x},\hat{y})-\beta_{\mathbf{x}}^*)_+^{\frac{1}{\gamma-1}}p_{\hat{Y}}(\hat{y}).$$

$\square$

## C. Convexity with regard to $\beta$

For simplciity, let $\zeta(\beta) = c_\gamma(\eta)\mathbb{E}_{\hat{y}\sim\mathbb{P}_{\hat{Y}}}\left[(h(\mathbf{x},\hat{y})-\beta)_+^{\gamma^*}\right]^{\frac{1}{\gamma^*}}+\beta$, then we have:

$$\begin{aligned}
&\zeta(\delta\beta_1 + (1-\delta)\beta_2) \\
&= \delta\beta_1 + (1-\delta)\beta_2 + c_\gamma(\eta)\mathbb{E}_{\hat{y}\sim\mathbb{P}_{\hat{Y}}}\left[\left(\delta(h(\mathbf{x},\hat{y})-\beta_1)+(1-\delta)(h(\mathbf{x},\hat{y})-\beta_2)\right)_+^{\gamma^*}\right]^{\frac{1}{\gamma^*}}
\end{aligned} \tag{29}$$

**Algorithm 1**

1: **Input:** Test-time input $\mathbf{x}$, ID labels $\mathcal{Y}_I = \{y_1, \ldots, y_K\}$, Negative labels $\hat{\mathcal{Y}} = \{\hat{y}_1, \ldots, \hat{y}_L\}$, learning rate $lr$, critic $h(\cdot, \cdot)$, maximum iteration $M$, hyper-parameters $\gamma^*, c_\gamma(\eta)$

2: **Output:** OOD scoring function $S_{\text{ours}}(\mathbf{x}; \boldsymbol{\theta})$

3: **repeat**

4:

$$\beta_{\mathbf{x}} \leftarrow \beta_{\mathbf{x}} - lr \cdot \frac{\partial}{\partial \beta_{\mathbf{x}}} \left\{ c_\gamma(\eta) \left[ \frac{1}{L} \sum_{j=1}^{L} \left( h(\mathbf{x}, \hat{y}_j) - \beta_{\mathbf{x}} \right)_+^{\gamma^*} \right]^{\frac{1}{\gamma^*}} + \beta_{\mathbf{x}} \right\}.$$

5: **until** convergence or reaching max iteration $M$

6: OOD scoring via

$$S_{\text{ours}}(\mathbf{x}; \boldsymbol{\theta}) = T \log \sum_{i=1}^{K} \frac{\exp \left\{ \frac{1}{T} \cdot [h(\mathbf{x}, y_i) - \beta_{\mathbf{x}}] \right\}}{\exp \left\{ \frac{c_\gamma(\eta)}{T} \cdot \left[ \frac{1}{L} \sum_{j=1}^{L} \left( h(\mathbf{x}, \hat{y}_j) - \beta_{\mathbf{x}} \right)_+^{\gamma^*} \right]^{\frac{1}{\gamma^*}} \right\}}$$

According to Minkowski's inequality, we have:

$$\mathbb{E}_{\hat{y} \sim \mathbb{P}_{\hat{Y}}} \left[ \left( \delta \cdot (h(\mathbf{x}, \hat{y}) - \beta_1) + (1 - \delta) \cdot (h(\mathbf{x}, \hat{y}) - \beta_2) \right)_+^{\gamma^*} \right]^{\frac{1}{\gamma^*}}$$

$$\leq \delta \cdot \mathbb{E}_{\hat{y} \sim \mathbb{P}_{\hat{Y}}} \left[ (h(\mathbf{x}, \hat{y}) - \beta_1)_+^{\gamma^*} \right]^{\frac{1}{\gamma^*}} + (1 - \delta) \cdot \mathbb{E}_{\hat{y} \sim \mathbb{P}_{\hat{Y}}} \left[ (h(\mathbf{x}, \hat{y}) - \beta_2)_+^{\gamma^*} \right]^{\frac{1}{\gamma^*}}, \tag{30}$$

such that

$$\zeta(\delta \beta_1 + (1 - \delta)\beta_2) \leq \delta \zeta(\beta_1) + (1 - \delta)\zeta(\beta_2). \tag{31}$$

## D. Algorithmic Summary

For clarity, we summarize our algorithmic details in Algorithm 1.

## E. Quantitative Analysis on Computation Time

As shown in Algorithm 1, our method consists of two stages: 1) Optimizing $\beta_{\mathbf{x}}$ and 2) OOD scoring.

As for the first stage, we start with deriving the specific form of the gradient w.r.t $\beta_{\mathbf{x}}$ as follows:

$$1 - c_\gamma(\eta) \phi(\beta)^{\frac{1}{\gamma^*} - 1} \frac{1}{L} \sum_{j=1}^{L} \mathbf{1} \left( h(\mathbf{x}, \hat{y}_j) > \beta \right) \left( h(\mathbf{x}, \hat{y}_j) - \beta \right)_+^{\gamma^* - 1},$$

where $\phi(\beta) = \frac{1}{L} \sum_{j=1}^{L} (h(\mathbf{x}, \hat{y}_j) - \beta)_+^{\gamma^*}$.

Clearly, the time complexity of gradient computation is $O(L)$, which, same as NegLabel, linearly grows with the number of negative labels. Here we omit the complexity introduced by the dot-product computation, as it is orthogonal to our algorithmic design. Therefore, the time complexity of the first stage which involve $M$-step SGD is $O(M \cdot L)$, where the maximum iteration $M$ is usually set to a relative small value (e.g., $M = 15$ in this paper)

As for the second step, one can easily check that the time complexity of our proposed scoring function $S_{\text{ours}}(\mathbf{x}; \boldsymbol{\theta})$ is $O(K + L)$, which is same as that of $S_{\text{NegLabel}}(\mathbf{x}; \boldsymbol{\theta})$ in Eq. (1).

Table 8 reports the average computation time of our method and NegLabel per test-time input on a single NVIDIA A100. Note that we omit the computation cost introduced by extracting features from pre-trained CLIP, as our method keeps the same feature extraction procedure as NegLabel.

Although the optimization process adds roughly 1.5 ms per test-time input, this cost is small in absolute terms. Therefore, both theoretically and empirically, the additional computation is limited and does not contradict our claim that the extra cost

*Table 8.* Time efficiency Analysis on the ImageNet benchmark.

| Ours (Stage 1) | Ours (Stage 2) | Ours (Stage 1+2) | NegLabel (with grouping strategy) |
|---|---|---|---|
| 1.51ms | 0.12ms | 1.63ms | 0.14ms |

*Table 9.* End-to-end latency analysis on the ImageNet benchmark.

| Ours (A100) | NegLabel (A100) | Ours (RTX8000) | NegLabel (RTX8000) |
|---|---|---|---|
| 116 images/s | 135 images/s | 101 images/s | 119 images/s |

is negligible in practice. To illustrate this, we further compare the end-to-end latency of our method with that of NegLabel across different GPU types (A100 and RTX8000), as shown in the table 9. We measure latency using frames per second (FPS)—that is, the average number of images processed per second. Note that a higher FPS indicates a lower latency.

# F. More Discussions

## F.1. Connection between $S_{\text{NegLabel}}(\mathbf{x}; f)$ and $\hat{S}_{\text{NegLabel}}(\mathbf{x}; f)$

For any $\alpha \in [0, 1]$, the $\alpha$-skew negative label distribution $\hat{\mathbb{P}}_{\hat{Y}}$ is defined as

$$\hat{\mathbb{P}}_{\hat{Y}} = \alpha \mathbb{P}_{Y_{\mathrm{I}}} + (1 - \alpha)\mathbb{P}_{\hat{Y}},$$

where $\mathbb{P}_{Y_{\mathrm{I}}}$ is the ID label distribution. Replacing $\hat{\mathbb{P}}_{\hat{Y}}$ with $\mathbb{P}_{\hat{Y}}$ in Eq. (7) gives

$$T \log \sum_{i=1}^{K} \frac{e^{h(\mathbf{x}, y_j)/T}}{\mathbb{E}_{\hat{y} \sim \hat{\mathbb{P}}_{\hat{Y}}}[e^{h(\mathbf{x}, \hat{y})/T}]} - T\eta$$

$$= T \log \sum_{i=1}^{K} \frac{e^{h(\mathbf{x}, y_j)/T}}{\alpha \mathbb{E}_{\hat{y} \sim \mathbb{P}_{Y_{\mathrm{I}}}}[e^{h(\mathbf{x}, \hat{y})/T}] + (1 - \alpha)\mathbb{E}_{\hat{y} \sim \mathbb{P}_{\hat{Y}}}[e^{h(\mathbf{x}, \hat{y})/T}]} - T\eta$$

$$\approx T \log \sum_{i=1}^{K} \frac{\exp[h(\mathbf{x}, y_i)/T]}{\frac{\alpha}{K} \sum_{j=1}^{K} \exp[h(\mathbf{x}, y_j)/T] + \frac{1-\alpha}{L} \sum_{j=1}^{L} \exp[h(\mathbf{x}, \hat{y}_j)/T]} - T\eta.$$

With $\alpha = \frac{K}{K+L}$, we have

$$
\begin{aligned}
&\log \sum_{i=1}^{K} \frac{\exp[h(\mathbf{x}, y_i)/T]}{\frac{\alpha}{K} \sum_{j=1}^{K} \exp[h(\mathbf{x}, y_j)/T] + \frac{1-\alpha}{L} \sum_{j=1}^{L} \exp[h(\mathbf{x}, \hat{y}_j)/T]} \\
&= \log \underbrace{\sum_{i=1}^{K} \frac{\exp[h(\mathbf{x}, y_i)/T]}{\sum_{j=1}^{K} \exp[h(\mathbf{x}, y_j)/T] + \sum_{j=1}^{L} \exp[h(\mathbf{x}, \hat{y}_j)/T]}}_{S_{\text{NegLabel}}(\mathbf{x}; f)} + \log(K + L)
\end{aligned}
\tag{32}
$$

The above implies that $S_{\text{NegLabel}}(\mathbf{x}; f)$ essentially estimates a worst-case energy function over a KL-divergence-constrained set, therefore being functionally equivalent to $\hat{S}_{\text{NegLabel}}(\mathbf{x}; f)$ in a broader sense. In a narrow sense, $S_{\text{NegLabel}}(\mathbf{x}; f)$ can be considered as a slightly noisy version of $\hat{S}_{\text{NegLabel}}(\mathbf{x}; f)$ (up to a constant) with the noise level $\alpha = \frac{1000}{1000+10000} \approx 0.09$.

### F.2. Comparison with Test-time Adaptation OOD Detection

We notice that, superficially, the use of 15-step SGD in test-time somewhat resembles the idea of test-time adaptation (TTA). However, we draw a conceptual distinction based on what is being updated and what the goal is:

TTA OOD DETECTION (E.G., ADANEG)

- **What is updated:** Negative proxies that are shared by test-time inputs.

*Table 10.* OOD detection results on ResNet-50 CLIP encoder under the OpenOOD setup, where ImageNet-1K is adopted as ID dataset.

| Near-/Far-OOD | Datasets | NegLabel | | Ours | |
|---|---|---|---|---|---|
| | | AUROC↑ | FPR95↓ | AUROC↑ | FPR95↓ |
| Near-OOD | SSB-hard | 65.51 | 88.19 | 66.82 | 86.72 |
| | NINCO | 73.87 | 73.58 | 76.39 | 71.47 |
| | Average | 69.69 | 80.89 | **71.60** | **79.09** |
| Far-OOD | iNaturalist | 99.26 | 2.80 | 99.37 | 2.36 |
| | Textures | 88.29 | 50.78 | 91.29 | 37.16 |
| | OpenImage-O | 91.80 | 33.73 | 91.67 | 34.86 |
| | Average | 93.12 | 29.10 | **94.11** | **24.94** |

- **Scope of update:** Updation is based on test-time inputs and can **accumulate over time**.

- **Goal:** Explicitly **adapt the negative proxies** to the test distribution.

- **Statefulness:** The negative proxies and/or their internal state after processing earlier test-time inputs **influence predictions on future test-time inputs**.

OUR METHOD

- **What is updated:** We **only optimize a scalar variable** $\beta_{\mathbf{x}}$ **per input**, which is *not* shared across test-time inputs.

- **Scope of update:** The optimization is **strictly local and input-dependent**. For each $\mathbf{x}$, $\beta_{\mathbf{x}}$ is reinitialized and optimized from scratch. There is no cross-sample sharing.

- **Goal:** We are **not** adapting negative labels to a new distribution. Instead, we are computing an **input-specific optimum** of a fixed energy function that was fully specified by the pre-trained CLIP model and the pre-computed negative labels.

- **Statefulness:** There is **no persistent state** that changes over time. OOD scoring for $\mathbf{x}_1$ and $\mathbf{x}_2$ are independent since optimizing $\beta_{\mathbf{x}_1}$ does not influence $\beta_{\mathbf{x}_2}$ or any future $\beta_{\mathbf{x}}$.

## G. More Experiment on ImageNet-1K

Our method is extensively evaluated against a range of OOD datasets under the OpenOOD setup (Zhang et al., 2023). Specifically, with ImageNet-1K as the ID dataset, we utilize Near-OOD datasets including SSB-hard (Vaze et al., 2021) and NINCO (Bitterwolf et al., 2023),and Far-OOD datasets including iNaturalist (Van Horn et al., 2018), Textures (Cimpoi et al., 2014), and OpenImage-O (Wang et al., 2022). As illustrated in Table 10, our advantage still holds.

