# OpenReview forum: "A Close Look at Negative Label Guided Out-of-distribution Detection in Pre-trained Vision-Language Models"
_ICML.cc/2026/Conference — ICML 2026 regular_

### Official Review · Reviewer_pVx4 · 2026-02-23

**Soundness:** 4
**Presentation:** 3
**Significance:** 4
**Originality:** 3
**Overall Recommendation:** 5
**Confidence:** 4

**Summary:**

The paper presents a zero-shot training free method to perform OOD detection using negative labels in vision language models such as CLIP. The paper uses uses concept of energy based scoring (a post-hoc method popular in single modal OOD detection) alongside provides theoretical analysis on selecting negative labels as worst case distribution over the set of an arbitrary distribution. The paper shows that by adopting such a formulation, the zero shot OOD detection performance can be significantly improved.

**Compliance With Llm Reviewing Policy:**

Affirmed.

**Key Questions For Authors:**

1. The NegLabel technique requires curating negative words from different external sources, how is the quality of those labels and their relevance to the downstream task ensured? Are such labels subjective to datasets used or are they generic? Also, regarding its practical relevancy, is there any additional computational constraint that comes along with it during testing?

2. How different is Q_Y_hat compared to P_Y_hat ? Is there any ablation conducted where performance across using these two distributions are compared? In addition, a closer look at their distribution manifold in 2D space as a visual repesentation would be more meaningful.

**Limitations:**

Yes

**Strengths And Weaknesses:**

Strengths:
The paper is technically sound where the claims are well supported by theoretical analysis and experimental results. The paper is clearly written and well structured with a good motivation of energy based OOD detection concept. Overall the paper is easy to follow. The paper addresses a well known problem in machine learning as the OOD problem is well researched and hot topic with high significance in the current trend. The paper is fairly original. The concepts used in the paper such as energy based method and negative label technique are already established and popular methods in the existing literature, however, the paper have done fair amount of work on relating two techniques as well as providing good theoretical analysis on them. The paper also have substantial amount of ablation studies performed across different architectures, datasets, and tasks.

Weakness:
The results section are not properly explained. For eg: there is no proper explanation of the key trends in the figures across different ablation. Also, there is no information on the standard error of the provided results which raises question for its statistical significance and the performance across multiple trials.

---

> ### Author Rebuttal · Authors · 2026-03-28
>
> # 1. The results section are not properly explained
> Thank you for pointing this out. We will carefully revise the manuscript by expanding the results section to provide a deeper analysis. In short, empirical trends explicitly highlight three takeaways: (i) the advantage over NegLabel is consistent across all tested backbones, indicating that the improvement is NOT architecture-specific; (ii) the same advantage persists at higher input resolution and across different corpus sources, suggesting robustness to both visual scale and negative-label source; and (iii) the hyperparameter analysis indicates a reasonably stable operating region NOT a sharply tuned optimum.
> # 2. Standard error of the provided results
> While this paper report averaged results over 5 independent runs, we, as per your constructive advice, are happy to include standard deviation as a complementary in the revised manuscript. Here, due to the space limitation, we report standard deviation of the provided results of Ours in Tables 2 and 3 as follows.
> |Table 2|iNaturalist (I)|Sun (S)|Places(P)|Textures (T)|
> |-|-|-|-|-|
> |FPR95|0.03|0.11|0.14|0.15|
> |AUROC|0.02|0.04|0.10|0.07|
>
> |Table 3 (ViT-B/32)|I|S|P|T|
> |-|-|-|-|-|
> |FPR95|0.04|0.08|0.11|0.20|
> |AUROC|0.05|0.03|0.08|0.06|
>
> |Table 3 (ViT-L/14)|I|S|P|T|
> |-|-|-|-|-|
> |FPR95|0.02|0.12|0.09|0.19|
> |AUROC|0.04|0.05|0.07|0.09|
>
> |Table 3 (ResNet50)|I|S|P|T|
> |-|-|-|-|-|
> |FPR95|0.05|0.16|0.08|0.12|
> |AUROC|0.02|0.07|0.05|0.06|
> # 3. How is the quality of negative labels and their relevance to the downstream task ensured?
> For a fair comparision, this paper uses negative labels that are filtered by  NegLabel. NegLabel does not simply pull random words; it uses a structured process to ensure the quality and relevance of negative labels:
> - NegLabel measures the distance between candidate negative labels and in-distribution (ID) labels. It selects labels with a "substantial margin" from the ID classes to ensure they represent truly different concepts.
> - By picking candidates with the greatest semantic distance from ID labels, the algorithm enhances the separability between ID and OOD labels.
>
> Theoretical analysis in Section 4 proves that NegLabel is inherently tolerant to sampling bias. It estimates a "worst-case" energy function over a distribution set centered on the negative labels, allowing it to perform reliably even when the curated labels deviate from real-world OOD data. However, the use of KL divergence in NegLabel makes NegLabel suffer from being overly pessimistic, which motivates our method.
> # 4. Are such labels subjective to datasets used or are they generic?
> The vocabulary source, e.g., WordNet, is generic, while the final selected negative labels are "subjective" only in the sense that they are filtered to ensure they do not overlap semantically with the specific ID classes of the task at hand.
>
> We argue that it is appropriate because OOD is defined relative to the downstream ID task. In the zero-shot setting, where true OOD labels are unavailable, this provides a principled proxy for enlarging the label space without requiring OOD supervision.
> # 5. Is there any additional computational constraint that comes along with it during testing?
> According to the original NegLabel paper, filtering $L=10,000$ negative labels from WordNet requires only a few minutes on a single NVIDIA RTX 3090 Ti GPU and can be conducted offline, resulting in minimal additional computational cost.
> # 6. Differenence between $\mathbb{Q}\_{\hat{Y}}$ and $\mathbb{P}\_{\hat{Y}}$
> In our paper, $\mathbb{P}\_{\hat{Y}}$ is the empirical negative-label sampling distribution while $\mathbb{Q}\_{\hat{Y}}$ is a worst-case distribution constrained within a divergence ball centered on $\mathbb{P}\_{\hat{Y}}$ (Eq. 5). Hence, the difference is **NOT** in label support but in how probability mass is reassigned. In particular, the KL-induced $\mathbb{Q}\_{\hat{Y}}$ results in an exponential reweighting toward highly similar negative labels while our proposed Rényi-induced $\mathbb{Q}\_{\hat{Y}}$ yields a milder polynomial reweighting that is less sensitive to outliers. This distinction is theoretically characterized in Theorems 4.3 and 5.2.
> # 7. Ablation and visualization
> We kindly note that Table 1, indeed, provides a study on this ablation: using the original $\mathbb{P}\_{\hat{Y}}$ (Eq. 4) gives 93.65 AUROC / 28.82 FPR95 on average; the KL-induced $\mathbb{Q}\_{\hat{Y}}$ (Eq. 7) gives 93.77 / 28.13; and our proposed Rényi-induced $\mathbb{Q}\_{\hat{Y}}$ (Eq. 11) improves further to 94.51 / 23.17.
>
> We agree that a 2D visualization can make this distinction more intuitive. The revised paper will include a t-SNE plot of the negative-label embeddings, with point weights under 1) $\mathbb{P}\_{\hat{Y}}$; 2) KL-induced $\mathbb{Q}\_{\hat{Y}}$; and 3) our proposed Rényi-induced $\mathbb{Q}\_{\hat{Y}}$.

---

> > ### Author Rebuttal · Reviewer_pVx4 · 2026-04-01
> >
> > Thank you authors for the rebuttal to my comments. My concerns have been fully addressed. However, I would like to remain consistent with my previous score just because of the reason cocnerning its originality and resemblance with NegLabel based methods.

---

> > > ### Author Response · Authors · 2026-04-02
> > >
> > > Thank you very much for your response and for confirming that all concerns have been addressed. Thank you again for your time and consideration.

---

### Official Review · Reviewer_PRXd · 2026-02-27

**Soundness:** 3
**Presentation:** 3
**Significance:** 3
**Originality:** 3
**Overall Recommendation:** 5
**Confidence:** 4

**Summary:**

This paper investigates the theoretical underpinnings of using "negative labels" (irrelevant class names) to enhance Out-of-Distribution  detection in pre-trained Vision-Language Models. The authors provide a mathematical analysis of the existing state-of-the-art method, NegLabel. They prove that NegLabel’s scoring mechanism is functionally equivalent to estimating an energy-based density function against a worst-case distribution constrained by a Kullback–Leibler (KL) divergence ball.
The authors identify a critical flaw in this reliance on KL divergence: it assigns exponentially high weights to "hard" negative labels (outliers), leading to over-pessimism and sensitivity to sampling bias. To address this, they propose a novel framework using Rényi divergence. The results achieve state-of-the-art performance on ImageNet-1K and OpenOOD benchmarks.

**Compliance With Llm Reviewing Policy:**

Affirmed.

**Final Justification:**

Concerns are addressed during rebuttal.

**Key Questions For Authors:**

N/A

**Limitations:**

yes

**Strengths And Weaknesses:**

Strengths:
1. This study re-frames heuristic negative label scoring into a formal energy-based density estimation framework. By proving the connection between NegLabel and Distributionally Robust Optimization (DRO) under KL constraints, the paper provides theoretical justification for why negative labels work.
2. The proposal to shift from KL divergence to Rényi divergence is well-motivated. The theoretical derivation showing how Rényi divergence leads to polynomial weighting effectively addresses the identified problem of outlier sensitivity.
3. The method demonstrates consistent improvements over strong baselines across a wide variety of setups.

Weaknesses:
1. Unlike standard post-hoc methods (like MCM or NegLabel) which are closed-form and instantaneous, the proposed method requires an iterative optimization process (SGD) for every test input to find the optimal parameter​. While the authors show in Appendix G that this only takes ~1.6ms and 15 steps, it technically moves the method from a pure inference-only calculation to a test-time optimization procedure.
2. While the method is robust to the distribution of negative labels, it still fundamentally relies on the existence of a large, diverse corpus of negative labels (e.g., WordNet). The performance is tied to the semantic gap between ID labels and the negative pool.

---

> ### Author Rebuttal · Authors · 2026-03-28
>
> We thank Reviewer PRXd for valuable comments. As to the questions and suggestions you raise, we took them seriously. Our response is as follows.
> # 1. Test-time optimization procedure.
> While our method is not closed-form in the same sense as MCM or NegLabel, we believe this lightweight per-sample optimization is well motivated from a **no-free-lunch perspective**: because the true test-time OOD distribution is unknown and may arbitrarily diverge from the sampled negative-label distribution, no single fixed closed-form scoring rule can be uniformly optimal across all such mismatch patterns.
>
> Our method addresses exactly this issue by replacing a fixed negative-label weighting with a small input-specific robust optimization that searches for a harder local worst-case weighting around the observed negative-label distribution, thereby reducing reliance on any one imperfect proxy for unseen OOD classes.
>
> Importantly, our method does **NOT** uses any ID training data and fine-tune the backbone, therefore remaining within the post-hoc/training-free zero-shot regime of CLIP-based OOD detection. On this basis, we argue the short SGD loop for optimizing a single scalar for each input should be viewed as **the computational price** of improved robustness to distribution mismatch, **NOT** as a departure from the post-hoc training-free setting.
> # 2. Reliance on the existence of a large, diverse corpus and the semantic gap between ID labels and the negative pool.
> We agree that performance can depend on the semantic relationship between the ID labels and the negative pool. However, such dependence is **NOT** a limitation unique to our method; it is **a fundamental challenge shared by essentially all negative-label-guided OOD detection methods**.
>
> Our contribution is **NOT to deny** this dependency, but to **improve robustness to it**. In particular, our work is motivated precisely by this open-world mismatch where real OOD semantics can arbitrarily deviate from the sampled negative labels. Rather than assuming the negative pool is perfectly aligned, we model such distribution discrepancy and derive a formulation that is more robust under mismatched or noisy negative labels.
>
> Thus, we argue that the key question should be how well it continues to perform when the negative pool is imperfect—which is the realistic regime our method targets. At the same time, we will dicussion such dependency in the Limitation section of the revised paper.

---

> > ### Author Rebuttal · Reviewer_PRXd · 2026-04-03
> >
> > Thanks for the reply, I will keep the score.

---

> > > ### Author Response · Authors · 2026-04-05
> > >
> > > Thank you very much for your response and for confirming that all concerns have been addressed. Thank you again for your time and consideration.

---

### Official Review · Reviewer_ctpA · 2026-03-12

**Soundness:** 3
**Presentation:** 3
**Significance:** 3
**Originality:** 3
**Overall Recommendation:** 5
**Confidence:** 4

**Summary:**

This paper studies zero-shot OOD detection with vision-language models using negative labels. Its main contribution is a robust optimization view of prior negative-label methods and a new divergence-based variant that is less sensitive to outliers and label mismatch. Experiments show consistent gains over strong baselines on OpenOOD benchmarks.

**Compliance With Llm Reviewing Policy:**

Affirmed.

**Final Justification:**

Interesting paper with strong and detailed rebuttal, I keep my positive score 5 and suggest an acceptance.

**Key Questions For Authors:**

- What is the runtime overhead versus NegLabel?

- Given the modest gain on ImageNet benchmark. How stable are the hyperparameter choices (e.g., $T, \gamma$) across different backbones and benchmark settings?

**Limitations:**

Limitation provided in the appendix, I would suggest move it to the main paper.

**Strengths And Weaknesses:**

$\textbf{Strengths:}$

- The paper is well motivated, as it targets an important and underexplored question: the theoretical foundation of NegLabel.

- The proposed analysis of OOD detection with negative labels through the lens of energy functions and density estimation is original and conceptually interesting.

- In addition, the paper includes extensive experiments on ImageNet-1K OOD benchmarks, domain-generalizable OOD settings, and multiple ablation studies, which together support the effectiveness of the approach.

$\textbf{Weaknesses:}$
My main concern is that the empirical gains, while consistent, are still modest on the main benchmark. On ImageNet-1K, the average improvement over NegLabel is about $+0.30$ AUROC and $-2.23$ FPR95, which is meaningful but not decisive given the huge amount of added theory and the per-input optimization of $\beta_x$.

---

> ### Author Rebuttal · Authors · 2026-03-29
>
> We appreciate the insightful comments provided by Reviewer ctpA. Please see our responses to your questions below.
>
> # 1. Empirical gains
> We sincerely thank for acknowledging the **consistency of our empirical gains**, and emphasize that the theoretical development is **NOT** meant only to explain a empirical gain.
>
> Prior to our work, there was a **limited** theoretical understanding of why negative labels are effective in open-world scenarios where true OOD data arbitrarily diverges from observed negative labels. Our theoretical framwork **mitigates this research gap** by formally explaining this inherent tolerance, and our proposed method is simply the natural, mathematically rigorous solution to the over-pessimism issue identified by this new framework.
>
> On this basis, we argue that the paper makes a meaningful and timely contribution by 1) offering a theoretical view of OOD detection with negative labels and 2) demonstrating consistent performance gains, where **the latter indeed justifies the generality of the former**.
> # 2. Runtime overhead versus NegLabel
>
> As this paper, following NegLabel, focuses on post-hoc OOD detection and therefore incurs no training cost.
>
> Compared with NegLabel, our method additionally requires a per-input optimization of $\beta\_x$. As detailed in Appendix G, optimizing $\beta\_x$ takes only **~1.5** milliseconds per image on an A100.
>
> Because the heavy feature extraction procedure of CLIP largely dominate the total runtime, our method (including feature extraction, optimizing $\beta_x$, and OOD scoring) achieves **116 images/sec** end-to-end on an A100, compared to NegLabel's **135 images/sec**. This minor **~14%** latency difference is effectively unnoticeable in real-world deployments.
>
> # 3. Hyperparameters
> As per your advice, we are happy to include more hyperparameter analysis on different backbones and benchmark settings as a complementary in the revised manuscript. Here, due to the space limitation, we report particial results as follows.
>
> | FPR95|iNaturalist|Sun|Places|Textures|
> |-|-|-|-|-|
> |ResNet50 ($T=0.008$) |1.54|24.97|41.96|43.05|
> |ResNet50 ($T=0.01$) |1.48|24.58|41.64|42.70|
> |ResNet50 ($T=0.02$)|1.65|25.37|42.35|43.43|
>
> | FPR95|iNaturalist|Sun|Places|Textures|
> |-|-|-|-|-|
> |ResNet50 ($\gamma=1.03$) |1.59|23.94|41.98|41.89|
> |ResNet50 ($\gamma=1.05$) |1.48|24.58|41.64|42.70|
> |ResNet50 ($\gamma=1.1$)|1.75|25.31|42.70|43.03|
>
> | FPR95|iNaturalist|Sun|Places|Textures|
> |-|-|-|-|-|
> |ImageNet-V2 ($T=0.008$) |1.21|22.64|41.62|43.41|
> |ImageNet-V2 ($T=0.01$)|1.35|23.20|42.05|44.77|
> |ImageNet-V2 ($T=0.02$) |1.42|23.94|42.98|45.53|
>
> | FPR95|iNaturalist|Sun|Places|Textures|
> |-|-|-|-|-|
> |ImageNet-V2 ($\gamma=1.03$) |1.69|23.94|41.99|45.05|
> |ImageNet-V2 ($\gamma=1.05$) |1.35|23.20|42.05|44.77|
> |ImageNet-V2 ($\gamma=1.1$)|1.48|24.11|42.35|45.63|
>
> # 4. Limitation is provided in the appendix
> We sincerely thank the reviewer for this constructive suggestion. In the revised paper, we have moved the limitation statement to the main content.

---

> > ### Author Rebuttal · Reviewer_ctpA · 2026-04-01
> >
> > Thanks for the detailed rebuttal. Given the high quality of the work, I will maintain my positive score.

---

> > > ### Author Response · Authors · 2026-04-02
> > >
> > > Thank you very much for your response and for confirming that all concerns have been addressed. Thank you again for your time and consideration.

---

### Official Review · Reviewer_6Upg · 2026-03-12

**Soundness:** 3
**Presentation:** 3
**Significance:** 3
**Originality:** 3
**Overall Recommendation:** 4
**Confidence:** 5

**Summary:**

This paper studies zero-shot out-of-distribution (OOD) detection with pretrained vision-language models, focusing on methods that use negative labels as auxiliary semantic anchors. The main goal is to explain why negative-label-based scoring works and to make it more robust when the sampled negative labels are imperfect proxies for true OOD semantics. The paper’s core contribution is a distributionally robust interpretation of negative-label OOD detection. It shows that existing negative-label scoring can be viewed through a worst-case optimization lens, where the model evaluates images against adversarial shifts around the negative-label distribution. Building on this view, the authors argue that the commonly induced KL-based worst-case weighting can be overly pessimistic because it may assign too much mass to a few highly similar negative labels. To address this, they replace the KL-style formulation, yielding a new score that reweights negative labels more smoothly and is intended to be less sensitive to noisy or misleading negatives. Empirically, the paper evaluates the proposed score across several CLIP backbones, prompt settings, negative-label corpora, and OOD benchmarks. The method is presented as a drop-in improvement over prior negative-label approaches and is also shown to be compatible with extensions such as adaptive negative-label selection and calibration modules. Overall, the paper contributes both a theoretical perspective on negative-label OOD detection and a modified scoring function that aims to improve robustness and performance in practice.

**Compliance With Llm Reviewing Policy:**

Affirmed.

**Final Justification:**

The rebuttal was well presented and addressed my concerns effectively. My overall opinion remains favorable, and I do not intend to revise my score.

**Key Questions For Authors:**

1. The connection between the original negative-label score and the KL-ball worst-case view should be clarified more carefully. Right now, this part seems to rely on several approximations and assumptions. Please clarify whether this is intended as a strict result or mainly as an interpretation.

2. The paper would be stronger with more direct evidence that the proposed score is indeed more robust to noisy negative labels.

3. Can you report variance across runs for the main results? Since many gains over prior methods are fairly small, it would be helpful to see standard deviations or confidence intervals.

**Limitations:**

The paper does not adequately discuss limitations or potential societal impact. The authors should briefly note dependence on negative-label quality, limited validation beyond CLIP, possible benchmark bias or contamination, and the extra test-time overhead.

**Strengths And Weaknesses:**

### Strengths
- One of the stronger aspects of the paper is that it tries to explain *why* negative-label methods work, instead of only proposing another score. The distributionally robust view gives a reasonable interpretation of existing methods and provides a basis for the new formulation.
- The paper is generally well organized, and the method is situated in the context of MCM, NegLabel, and follow-up work, so readers can understand the intended contribution.
- Although the paper is not proposing a new paradigm, improvements in this training-free setting are still useful. Methods of this kind are attractive because they are easy to plug into existing pipelines.

### Weaknesses
- My main concern is that the strongest theoretical claims rely on approximations and assumptions that are not really stress-tested in the experiments. The connection between the original negative-label score and the KL-ball worst-case view is interesting, but it does not feel fully airtight in the way the paper sometimes suggests.
- The paper argues that KL-based weighting can be overly pessimistic and too sensitive to a few highly similar negative labels, but this point is not examined as directly as it could be. A more convincing analysis would explicitly add noisy negatives, subclass labels, or false negatives and then compare the different weighting schemes under controlled settings.
- The empirical gains are generally positive but not large. Because the margins over prior methods are often small, it is hard to judge how robust those gains are without standard deviations, confidence intervals, or some measure of run-to-run variation.
- The work is original more in interpretation than in overall design. That is still valuable, but I do not see it as a major methodological leap.
- In practice, the method remains very close to NegLabel and mainly changes how negative evidence is aggregated. So the novelty is real, but it is best described as a principled refinement rather than a fundamentally new approach.

---

> ### Author Rebuttal · Authors · 2026-03-30
>
> # 1. Ablation on negative labels
> Thanks for this insightful advice. To give direct evidence, we conduct experiments on the ImageNet-1K benchmark where we use **randomly selected words** from WordNet as a negative label set that is inevitably noisy. The table below shows that ours still outperforms NegLabel.
> |FPR95/AUROC|iNaturalist (I)|Sun (S)|Places(P)|Textures (T)|
> |-|-|-|-|-|
> |NegLabel|9.11/97.96|28.67/93.93|45.10/89.54|55.87/86.62|
> |Ours |7.60/98.40|24.78/94.91|38.59/92.23|50.26/89.42|
> # 2. Standard deviations
> While this paper report averaged results over 5 independent runs, we, as per your constructive advice, are happy to include standard deviation as a complementary in the revised manuscript. Here, due to the space limitation, we report standard deviation of the provided results of Ours in Tables 2 and 3 as follows.
> |Table 2|I|S|P|T|
> |-|-|-|-|-|
> |FPR95|0.03|0.11|0.14|0.15|
> |AUROC|0.02|0.04|0.10|0.07|
>
> |Table 3 (ViT-B/32)|I|S|P|T|
> |-|-|-|-|-|
> |FPR95|0.04|0.08|0.11|0.20|
> |AUROC|0.05|0.03|0.08|0.06|
>
> |Table 3 (ViT-L/14)|I|S|P|T|
> |-|-|-|-|-|
> |FPR95|0.02|0.12|0.09|0.19|
> |AUROC|0.04|0.05|0.07|0.09|
>
> |Table 3 (ResNet50)|I|S|P|T|
> |-|-|-|-|-|
> |FPR95|0.05|0.16|0.08|0.12|
> |AUROC|0.02|0.07|0.05|0.06|
> # 3. Originality
> Thanks for acknowledging the **valuable originality** of our theoretical interpretation. We respectfully argue that our methodological contribution is inextricably linked to and elevated by this new interpretation, representing a highly principled methodological leap.
> - Our methodological design is **NOT** derived from empirical trial and error, **but** rather as a direct solution to the specific mathematical flaw we identified in existing methods. By proving that NegLabel implicitly estimate energy against a worst-case distribution within a KL-divergence ball, we uncovered the root cause of their over-pessimism and sensitivity to outliers
> - The shift to utilizing the Cressie-Read family of Rényi divergence is a significant methodological leap. Unlike prior heuristic tricks, introducing the order parameter $\gamma$ provides a theoretically grounded mechanism to flatten the effect of outliers, transforming the exponential weights of KL divergence into milder, polynomial-bounded weights
> # 4. Novelty
> Thanks for acknowledging the **real novelty and principled nature** of our method. While our method shares similarities with NegLabel by utilizing a set of negative labels, we respectfully highlight that this "principled refinement" represents a fundamental shift from empirical heuristics to a rigorous theoretical foundation.
> - Our work provides the first mathematical explanation for why this NegLabel in open-world scenarios by formally proving (Theorems 4.1 and 4.2) that NegLabel implicitly estimates a worst-case energy function over a KL-constrained set
> - Our theoretical framework enables to mathematically identify a critical, previously unnoticed flaw in NegLabel: the implicit KL-divergence constraint induces excessive pessimism (Theorem 4.3)
> - Our transition to Rényi divergence is **NOT** merely a different aggregation choice, **but** a theoretically derived solution to flatten the effect of these outliers (Theorems 5.1 and 5.2). As demonstrated in our extensive experiments, resolving this fundamental flaw yields a highly robust framework that establishes a new state-of-the-art across diverse settings
> # 5. Connection between the NegLabel and the KL-ball worst-case view
> Thanks for pointing this out. We agree that the current presentation does not sufficiently distinguish the exact KL-robust result from the approximate connection to the original NegLabel score.  In the revised paper, we will add a dedicated paragraph to clearly summarize the following points.
> - the KL-ball worst-case formulation is a strict result for the robustified energy in Eq. (5), as established by Theorems 4.1 and 4.3
> - the connection from this robustified energy to $\hat{S}\_{\text{NegLabel}}$ in Eq. (7), and subsequently to the original $S\_{\text{NegLabel}}$, should be viewed mainly as an interpretation since it relies on the approximation of $\alpha^\*(x, P\_{\hat{Y}})$, the homoscedasticity assumption used to relate $\alpha^\*$ to $T$, and the empirical approximation of the expectation using sampled negative labels
> - Appendix H.1 provides a broader functional equivalence between $\hat{S}\_{\text{NegLabel}}$ and $S\_{\text{NegLabel}}$ through an $\alpha$-skew mixture construction, rather than an exact algebraic identity for the original NegLabel objective
> # 6. The paper does not adequately discuss limitations or potential societal impact.
> Thanks for your valuable comments. We completely agree that our paper can be improved by discussing limitations and broader impacts more explicitly. In the revised version, we will expand the Limitations and Societal Impact section to carefully discuss these raised insightful factors.

---

> > ### Author Rebuttal · Reviewer_6Upg · 2026-04-01
> >
> > The rebuttal was clear and helpful. My overall assessment stays positive, so I do not plan to change my score.

---

> > > ### Author Response · Authors · 2026-04-02
> > >
> > > Thank you very much for your response and for confirming that all concerns have been addressed. Thank you again for your time and consideration.

---

### Decision · Program_Chairs · 2026-04-30

**Decision:**

Accept (regular)

**Comment:**

This paper studies zero-shot OOD detection for vision-language models and aims to better understand why NegLabel-based methods, which rely on imperfect proxies of true OOD classes, can nevertheless be effective. The authors provide an interpretation that connects negative-label formulations to a worst-case optimization perspective, and further propose a Rényi-divergence-based energy method as an alternative to KL-based weighting, which is argued to be overly conservative. The proposed method can be combined with existing approaches and demonstrates improved performance.

In the initial review round, reviewers generally recognized several strengths of the work, including the novel interpretation of negative labels (6Upg, PRXd), the originality of the problem and idea (ctpA), the soundness of the proposed formulation (PRXd), empirical effectiveness (6Upg, ctpA, PRXd), and overall clarity (6Upg).

At the same time, reviewers raised several concerns. These included the need for test-time optimization rather than a purely closed-form inference procedure (PRXd), reliance on theoretical assumptions or approximations that are not fully validated empirically (6Upg), relatively modest performance gains (6Upg, ctpA), limited methodological novelty (6Upg), dependence on external corpora (PRXd), and insufficient analysis of computational cost and hyperparameter sensitivity (ctpA).

The authors provided detailed responses addressing these concerns, and the reviewers generally acknowledged the clarifications. While some points remain partially open, such as the dependence on the quality of the external corpus, and a remaining gap in the theoretical argument connecting NegLabel and the KL-ball worst-case formulation (in particular the approximation between $S_{\text{NegLabel}}$ and $\hat{S}_{\text{NegLabel}}$), these aspects are at least partially quantified and supported by empirical evidence.

Overall, the paper offers a useful theoretical perspective on NegLabel-based OOD detection and complements it with consistent empirical improvements. Despite some remaining limitations, the contributions are considered meaningful in both conceptual and practical aspects.